# Microfluidic space coding for multiplexed nucleic acid detection via CRISPR-Cas12a and recombinase polymerase amplification

Zhichen Xu[1,2,5], Dongjuan Chen[3,5], Tao Li[1,2], Jiayu Yan[1,4], Jiang Zhu[1,2], Ting He[1,2], Rui Hu[1,2], Ying Li [1,2] ✉, Yunhuang Yang[1,2] & Maili Liu [1,2]

Fast, inexpensive, and multiplexed detection of multiple nucleic acids is of great importance to human health, yet it still represents a significant challenge. Herein, we propose a nucleic acid testing platform, named MiCaR, which couples a microfluidic device with CRISPR-Cas12a and multiplex recombinase polymerase amplification. With only one fluorescence probe, MiCaR can simultaneously test up to 30 nucleic acid targets through microfluidic space coding. The detection limit achieves 0.26 attomole, and the multiplexed assay takes only 40 min. We demonstrate the utility of MiCaR by efficiently detecting the nine HPV subtypes targeted by the 9-valent HPV vaccine, showing a sensitivity of 97.8% and specificity of 98.1% in the testing of 100 patient samples at risk for HPV infection. Additionally, we also show the generalizability of our approach by successfully testing eight of the most clinically relevant respiratory viruses. We anticipate this effective, undecorated and versatile platform to be widely used in multiplexed nucleic acid detection.

Fast, inexpensive, and multiplexed detection of pathogens is of great value to human health and global security[1,2]. Several diagnostic applications require the rapid detection of diverse nucleic acids. These include the identification of different types of pathogens (e.g., the quick differentiation of SAR-CoV-2 from the influenza virus or other respiratory viruses during the COVID-19 pandemic)[3-6] and the discrimination of viral variants (e.g., subtyping of human papillomavirus [HPV], which has >100 variants)[7,8]. Next-generation sequencing (NGS) can provide detailed information and allow the identification of a wide range of pathogens[9,10]. However, the turnaround time and cost are both high[7]. Although various nucleic acid amplification methods can be used to detect multiple targets at a lower cost, some limitations and challenges remain[11]. An ideal multiplexed detection platform would have the capability to address these above-mentioned problems. Such a platform should be able to:

(i) correctly and efficiently amplify multiple (e.g., ≥4) nucleic acid targets; (ii) amplify and identify the targets at a suitable temperature; (iii) precisely recognize the targets without any interference; and (iv) accurately distinguish between signals and correlate them with each individual target[12,13].

Polymerase chain reaction (PCR) has long been utilized for multiplexed amplification. Real-time PCR is commonly used as the gold standard method for pathogen identification[4,14]. To examine multiple targets, real-time PCR often relies on the detection of different fluorophores during amplification through nucleic acid hybridization[6,14,15]. However, this detection approach has two primary limitations. First, the emission spectra of fluorescent probes often overlap, and therefore, the number of targets that can be tested is limited. Additionally, off-target amplification can confound target identification owing to the unavailability of rectification

[1]State Key Laboratory of Magnetic Resonance and Atomic Molecular Physics, National Centre for Magnetic Resonance in Wuhan, Wuhan Institute of Physics and Mathematics, Innovation Academy for Precision Measurement Science and Technology—Wuhan National Laboratory for Optoelectronics, Chinese Academy of Sciences, Wuhan 430071, China. [2]University of Chinese Academy of Sciences, Beijing 10049, China. [3]Department of Laboratory Medicine, Maternal and Child Health Hospital of Hubei Province, Tongji Medical College, Huazhong University of Science and Technology, Wuhan 430070, China. [4]School of Physical Education, China University of Geosciences, Wuhan 430074, China. [5]These authors contributed equally: Zhichen Xu, Dongjuan Chen. ✉e-mail: liying@wipm.ac.cn

mechanisms[7], and this problem worsens during multiplexed detection. Thus, although real-time PCR can address the first two challenges, it does not solve the last two.

Recently, CRISPR-based diagnostic (CRISPR-Dx) methods have been widely implemented for diagnosing infections due to their high specificity and sensitivity[3,12,16–18]. These platforms harness the collateral cleavage activity of various Cas proteins (e.g., Cas12a[19,20], Cas13a[21,22], Cas14[6,23]) to cut the designed reporters after activation. To further increase detection sensitivity, CRISPR-powered detection strategies are typically combined with isothermal amplification methods such as recombinase polymerase amplification (RPA)[1,24,25] and loop-mediated isothermal amplification (LAMP)[3]. RPA, a fast and high-fidelity amplification method, is used more commonly because the reaction requires only a single temperature (37–42 °C) that can be easily achieved with simple heaters and is also compatible with the optimum temperature of CRISPR-Cas systems[13,26–28]. Moreover, the detection specificity is guaranteed due to the sequence specificity between CRISPR RNA (crRNA) and the amplicons[1,16,28].

The first CRISPR-Dx platform called SHERLOCK was developed using Cas13a and reverse-transcription RPA (RT-RPA) for the detection of Zika and Dengue viruses. This platform had attomole (aM) sensitivity and single-base mismatch specificity[21]. Subsequently, the multiplexed platform SHERLOCKv2 was developed by combining four CRISPR enzymes (LwaCas13a, PsmCas13b, CcaCas13b, and AsCas12a) with specific cleavage preferences and four types of fluorescent probes (FAM, TEX, Cy5, and HEX) to detect four targets[1]. This platform had high potential as a multiplexable and sensitive platform for nucleic acid testing (NAT). However, it was challenging to increase the number of targets because of the difficulties in choosing additional appropriate CRISPR enzymes and probes[16,29]. Therefore, in order to identify more pathogen targets, another Cas13-based platform called CARMEN was developed recently[7,30]. This system allows the simultaneous detection of up to 169 human-associated viruses using microwell array technology to pair CRISPR reagents and amplified samples. The readouts are obtained by identifying a pool of color codes that result from different ratios of four Alexa Fluor dyes. Though CARMEN represents a milestone for scalable and multiplexed pathogen detection, its widespread use may be compromised due to the multi-step color coding/decoding and the complicated assay protocol[30]. Additionally, owing to the four-fluorophore-based color-coding strategy, concerns regarding signal overlap due to the wide emission spectra of the fluorophores remain. Therefore, despite advancements in CRISPR-Dx technologies for multiplexed pathogen detection, the fourth challenge (generating distinguishable signal codes for multiple targets) remains unmet. Indeed, a recent work developed an improved CARMEN system (named microfluidic CARMEN, mCARMEN), which relies on commercially available Fluidigm microfluidics and instrumentation[30]. It could be a promising strategy for the detection of multiple viruses, but the instrumentation setup and the device are relatively expensive and these might compromise its wide application in resource-restricted areas. Therefore, an "ideal" suitable and versatile platform that fulfills all the four requirements remains elusive.

Herein, we report the development of a unique platform that couples a microfluidic device with CRISPR-Cas12a and multiplex RPA (MiCaR) to address all the challenges associated with multiplexed NAT. In addition to deploying the promising features of RPA and the CRISPR system, our strategy utilizes the unique properties of microfluidics to allow precise fluid control and spatial reagent arrangement. Instead of using multiple fluorophores or color coding to indicate different targets, MiCaR distinguishes between target signals based on spatial positions or space coding using only a single fluorophore. By pre-loading various crRNAs into the 30 designated wells, the system simultaneously initiates up to 30 CRISPR-based cleavage assays after the sample is pipetted onto the center of the Starburst-Shaped microchip (SS-Chip).

In this study, we evaluate the performance of the SS-Chip to ensure homogenous liquid division and mixing. We select the nine HPV subtypes targeted by 9-valent HPV vaccine (9vHPV). Subsequently, we carefully optimize and demonstrate the efficiency and specificity of the multiplex RPA assay. Additionally, we conduct a test containing 9 × 9 crRNAs against HPV subtypes to determine the optimal crRNA combination. Moreover, extensive testing demonstrates that MiCaR can achieve a detection sensitivity of 0.1 nM and 0.26 aM for unamplified and amplified targets, respectively. Furthermore, we apply MiCaR to test samples from patients at risk of HPV infection. In total, we test 100 samples, and 3000 assays were performed. MiCaR shows great stability, and the on-chip HPV subtyping results are highly consistent with clinical results, achieving a sensitivity of 97.8% and specificity of 98.1%. Finally, we apply MiCaR to successfully test eight common respiratory viruses to confirm the generalizability of our approach. Therefore, the findings show that MiCaR allows rapid, low-cost, and multiplexed NAT with high sensitivity, specificity, and reliability. This approach could be widely applied for the diagnosis of specific infections and provide significant health-related and clinical benefits.

## Results
### Principle of the MiCaR platform

To address all the four challenges mentioned earlier, we developed the MiCaR system (Fig. 1) as a universal platform for multiplexed NAT. Herein, we demonstrate its use in HPV subtyping. After sampling, cervical cells were heated to induce the release of HPV DNA. Then, direct multiplex RPA was performed without DNA extraction (Fig. 1a). The amplified products were loaded onto the microfluidic device and tested using the CRISPR-based cleavage assay. The readout was obtained via an automatic fluorescence imaging system. As shown in Fig. 1b, the detection by MiCaR relies on a Starburst-Shaped Chip (SS-Chip), which is designed as a hub-and-spoke network. The sample is loaded into the hub well at the center, and the spoke microchannels homogeneously distribute the sample into 30 labeled wells pre-loaded with a specific Cas12a-detection mix containing Cas12a, crRNA, and a fluorescence reporter. The reporter used is a fluorophore quencher (FQ)-labeled oligonucleotide named TBA11 (GGTTGGTGTGG), a truncated form of thrombin binding aptamer (TBA), that has been proven to have higher sensitivity than a normal ssDNA reporter[31,32]. If the HPV DNA matches the crRNA, the relevant wells (three wells used as a group for testing the same HPV subtype) will show a bright fluorescence signal. However, a low background signal will be obtained when there is a mismatch between the HPV DNA and the crRNA. A blank control (with reporter and Cas12a in the absence of crRNA) was tested on the device to obtain the basic background. This space-coding based strategy enabled the efficient identification of multiple targets by employing the "amplify together and detect individually" principle. The use of only one fluorescent probe eases system setup and data analysis, and the triplicate on-chip test (for the same target) also ensures detection accuracy and reliability.

### Primer and crRNA design for the HPV panel

There are >100 HPV subtypes, including high-risk and low-risk types, which can cause cervical cancer and condyloma acuminatum, respectively[8]. As the risk associated with different HPV subtypes varies, it is necessary to screen and identify infections and prevent subsequent complications, such as cervical cancer. Vaccines can prevent cancer and diseases caused by some HPV infections. Currently, 9-valent (HPV−6, −11, −16, −18, −31, −33, −45, −52, −58) HPV vaccines are considered the most powerful for preventing HPV infection. In this study, we selected the nine HPV subtypes targeted by the 9vHPV to develop the MiCaR-based detection system. We first designed the relevant RPA primers for the nine targets. RPA is a highly sensitive isothermal amplification method that requires minimal sample

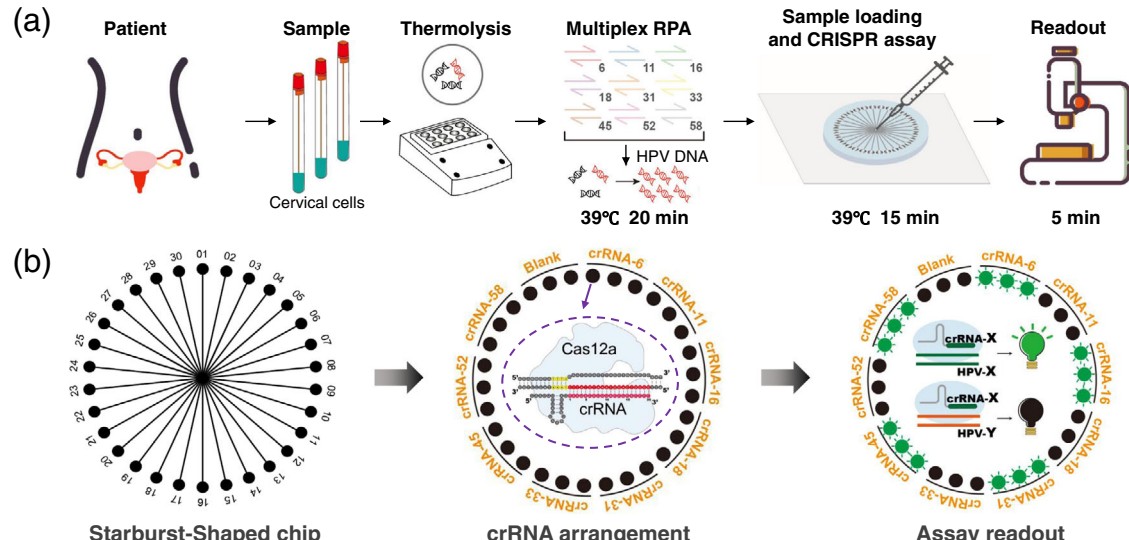

**Fig. 1 | Scheme of the MiCaR-based HPV subtyping strategy. a** Brief overview of the steps involved in the subtyping process. The collected cervical cell specimen is first thermolyzed. Then, RPA is performed to amplify the nine HPV subtypes. The amplicons are subsequently tested via the CRISPR-Cas12a system on the microfluidic device, followed by fluorescence imaging to obtain the readout. **b** On-chip testing principles. A 30-plexed starburst-shaped chip (SS-Chip) with one central inlet connected to 30 outlets is used. The outlets are preloaded with various Cas12a/crRNAs that recognize the relevant target HPV subtype. After the on-chip assay, the fluorescent readout at specific outlets (i.e., space coding) indicates the presence of relevant HPV subtypes in the sample.

preparation (Fig. 2a, left). Since the HPV subtype is determined by the L1 gene sequence, we designed a series of primers against this region in the nine targets (Fig. 2a, right).

The L1 gene sequences of these nine HPV subtypes show high similarity (Supplementary Data 1, Supplementary Table 1). To select the applicable primers, two rounds of designing and testing were performed. In round one, 10 pairs of primers were designed (Supplementary Table 2, one for each HPV subtype, with an additional one for HPV-11 owing to its high similarity with HPV-6; Supplementary Table 1). The performance of these primers was determined using agarose gel electrophoresis (Supplementary Fig. S1a). Most of the primers worked well and displayed clear and dense bands. However, primers for HPV-6 and HPV-45 did not generate distinct bands. Therefore, three additional pairs of primers were designed for these two subtypes based on the general principles of primer design (Supplementary Table 3). It was observed that the Primer HPV-6a, -6b, -45a, -45b, and -45c amplified the relevant targets successfully (Supplementary Fig. 1b).

To choose the preferable primers for HPV-6, -11 and 45, we comprehensively checked the sequence similarity between the amplification regions of the primer candidates (HPV-6a, -6b, -11a, -11b, -45a, -45b and -45c) and the L1 gene of the other eight subtypes. As shown in Supplementary Fig. 2a and Supplementary Data 2, the amplification region of Primer 6b showed a relatively greater difference than Primer 6a, which should contribute to reducing potential cross-reactivity in the subsequent CRISPR-based assay. Similarly, Primer 11b and 45a were theoretically superior to their counterparts (Supplementary Data 2, Supplementary Table 4). Therefore, Primer HPV-6b, -11b, -16, -18, -31, -33, -45a, -52, -58 were selected for the final panel (Supplementary Table 5). The gel results demonstrated the successful amplification of all nine targets (Fig. 2b). The L1 gene sequences and amplification regions of the nine HPV subtypes are listed in Supplementary Data 3.

In addition to an appropriate amplification region, optimal primer constitutions and concentrations are important for successful multiplex RPA assays. The Basic Kit Quick Guide for single RPA recommends a primer concentration of 1 μM. Thus, we initially tested RPA performance with the nine targets and nine pairs of primers (1 μM for each pair, and 9 μM in total). As shown in Supplementary Fig. 3a, severe primer dimer formation was observed, and no amplicons were obtained. Hence, RPA assays were conducted using only one target

(HPV-16) with lower primer concentrations (0.1, 1, and 5 μM in total) to identify the best primer pool. As indicated in Supplementary Fig. 3b, the pool with a total concentration of 1 μM (~0.11 μM for each pair) appeared optimal for efficiently amplifying the target HPV-16 without causing significant primer dimer formation. Hence, this primer constitution was used for subsequent assays.

Next, we evaluated the performance of the primer pool in multiplex RPA. The products of the 9-plexed PRA were identified through next-generation sequencing (NGS) (Supplementary Fig. 4). The results indicated that a variety of amplicons were produced for each of the nine targets (Fig. 2c). This was expected, as amplicons of different lengths can be generated within a reaction for a single target in RPA. The top 5 sequences of the products showed high-fidelity to the original sequences (Supplementary Data 4). The relatively lower amplicon fidelity of HPV-11 and HPV-18 was resulting from one or two amplicons with shorter lengths comparing to the original templates. Anyway, all these amplicons showed a 100% fidelity in the crRNA binding regions. These results demonstrated the effectiveness of the primer pool.

Subsequently, we tried to optimize the crRNA pool. Initially, the crRNA pool was generated by only comparing the crRNA of a certain target to the amplification regions of non-targets (Supplementary Table 6). However, this led to cross-activity between the initial crRNA of HPV-16 (HPV-6 crRNA-I) and HPV-11 (Supplementary Fig. 5). To efficiently avoid cross-reactivity among the targets, a refined approach was used to design the optimal crRNA pool. HPV-16 was selected to generate proof of concept. First, all four potential crRNAs were designed within the RPA region according to the basic principles of crRNA design (Fig. 2d). Then, each crRNA was compared with the full L1 gene (and not just the amplification region) of the other eight HPV subtypes (Supplementary Fig. 6). Given that HPV is a double-stranded DNA virus, the sequence comparison could be performed directly using the crRNA and gene sequence. Comparison results showed that crRNA-2, -3, -4, and -5 were highly similar (difference of only two nucleotides) to certain regions of the full L1 gene in the other eight HPV subtypes (Fig. 2d, Supplementary Fig. 6b–d). Accordingly, crRNA-1 that showed the most significant differences was selected for HPV-16 (Supplementary Fig. 6a). The crRNAs for the other subtypes were designed in a similar manner. All the crRNA sequences are listed in Supplementary Table 7.

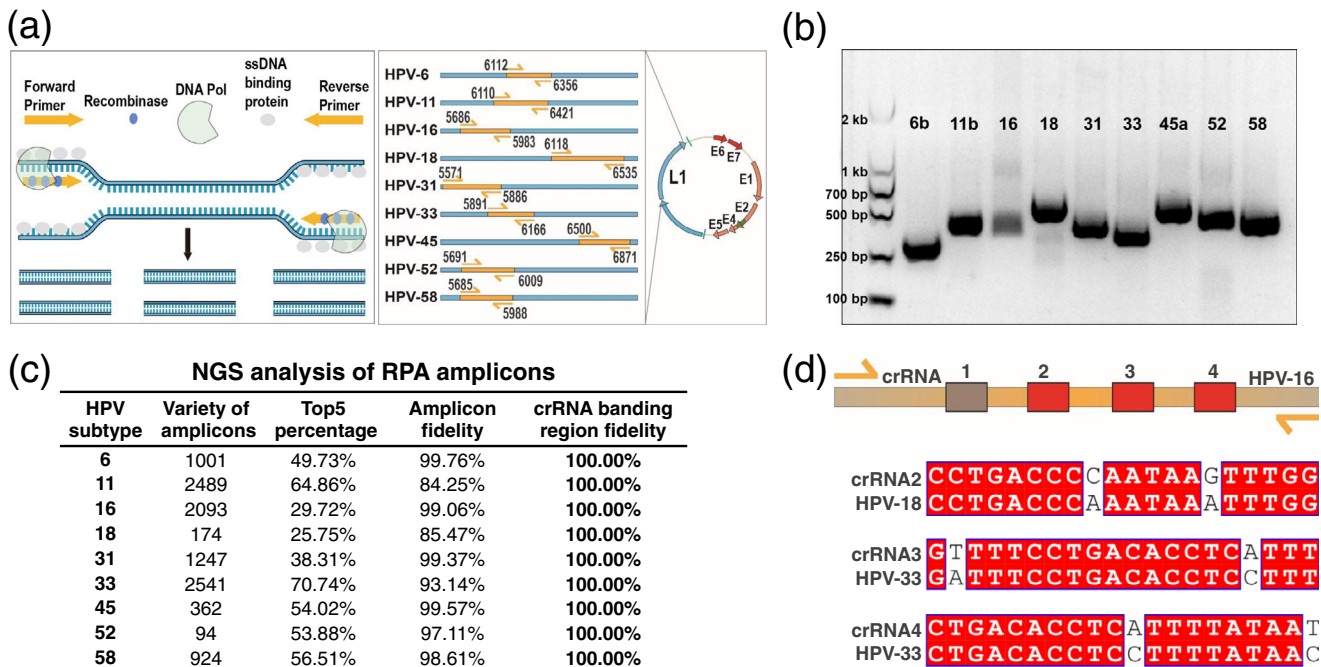

**Fig. 2 | Designing of RPA primers and crRNAs. a** RPA primer designing for the nine HPV subtypes. **b** Agarose gel results showing the RPA products of the nine targets after amplification with optimal primers. The experiment was repeated at least two times independently. **c** Identification of the 9-plexed RPA products via NGS. **d** Representative Cas12a-associated crRNA designing for HPV-16.

## Validation of crRNAs and RPA primers for the nine HPV subtypes

Using the comprehensive theoretical analysis described above, optimal crRNAs were designed for all nine HPV targets. Then, a 9 × 9 matrix test (9 crRNAs × 9 HPV L1 gene plasmids; Fig. 3a, left panel) was designed to verify crRNA performance (i.e., recognition specificity). Additionally, a control group without any target gene was set up. After testing, widespread cross-activity was not observed (Fig. 3a, right panel). Similar trends were observed when the crRNAs were analyzed separately (Fig. 3b). Of note, the crRNA for HPV-58 showed a relatively high background for HPV-33. However, this was negligible compared to the signal for the actual matched sample (crRNA [HPV-58] against HPV-58). These crRNAs with high specificity were then used for subsequent multiplexed NAT.

We next applied the CRISPR/Cas12a system to detect the products obtained through the 4-plexed and 9-plexed RPA assay. The trans-cleavage capability of Cas12a is activated when the crRNA recognizes its HPV target in the RPA product. Accordingly, the TBA11 reporter is cut into smaller elements. The detection results were first analyzed using denaturing PAGE. Figure 3c shows that the reporters in lanes 2–5 had been cleaved. In the other lanes, the reporter was intact, indicating that the Cas12a system was not activated. Next, the 9-plexed RPA products were evaluated. As shown in Fig. 3d, the results demonstrate that all nine lanes (lanes 2–10) had small reporter segments, suggesting that the nine HPV subtypes were efficiently amplified and successfully activated the Cas12a system. Some bands corresponding to cleaved reporters had different intensities (Fig. 3c, d), which could be due to variations in the cleavage efficiencies of the Cas12a system activated by different targets. The control (no HPV template) did not activate Cas12a in either the 4-plexed or 9-plexed assay.

Next, we examined the cleavage products of the 4-plexed and 9-plexed assays based on the fluorescence readout that would be used in subsequent chip-based experiments. Consistent with the results of denaturing PAGE, four and nine bright fluorescence signals were observed in the 4-plexed and 9-plexed assays, respectively (Fig. 3e). These results confirmed the excellent performance of the multiplex RPA and CRISPR/Cas system, and laid a solid foundation for the subsequent on-chip HPV subtyping analysis. Based on these results, we also proposed a procedure for designing RPA primers and crRNAs for multiplexed NAT (Fig. 3f).

Furthermore, we investigated the detection sensitivity of the multiplex RPA. A series of plasmid samples were prepared for the nine HPV targets with different concentrations ($10^{-12}$, $10^{-13}$, $10^{-14}$, $10^{-15}$, $10^{-16}$, $10^{-17}$, $10^{-18}$ and 0 M). These samples were amplified and measured via the Cas12a-based assay (Fig. 4a). The detection results were shown in Fig. 4b. The 9-plexed RPA assay (1× primer for each target) achieved a sensitivity of $10^{-17}$–$10^{-18}$ M for all the targets except HPV-18. The lower sensitivity for HPV18 could result from a relatively lower primer efficiency during the amplification for low-concentration templates. This was improved by using 3× HPV18 primer in the subsequent 9-plexed RPA assay, and finally it also approached to $10^{-18}$ M.

## Characterization of the SS-Chip and MiCaR detection system

We conducted a series of experiments to evaluate the performance of the SS-Chip. Micrographs and photographs of the SS-Chip, which has a microwell radius of 750 μm and a microchannel width of 100 μm, are shown in Fig. 5a and Supplementary Fig. 7. The SS-Chip could deliver the fluid from the central well to the surrounding 30 outlet wells through its microchannels. To assess the aliquoting ability of the SS-Chip, 240 μL of food dye was injected into the central well. The volume of dye in each surrounding well was assessed using a precision Hamilton syringe. As shown in Fig. 5b, the solution could be divided equally within the 30 designated wells. The operation of the SS-Chip is shown in Supplementary Movie 1.

To mimic the solution mixing occurring during the cleavage reaction, a fluorescein solution was pre-loaded into the 30 designated wells. Then, sulforhodamine B solution was injected into the central well. After 5 min, fluorescence images of the green and red channel were captured for each well (Fig. 5c). The merged image showed a homogeneous yellow color in each well, demonstrating that the solutions were completely mixed. Then, a Cas12a cleavage assay was conducted using the device to verify whether the CRISPR/Cas12a detection

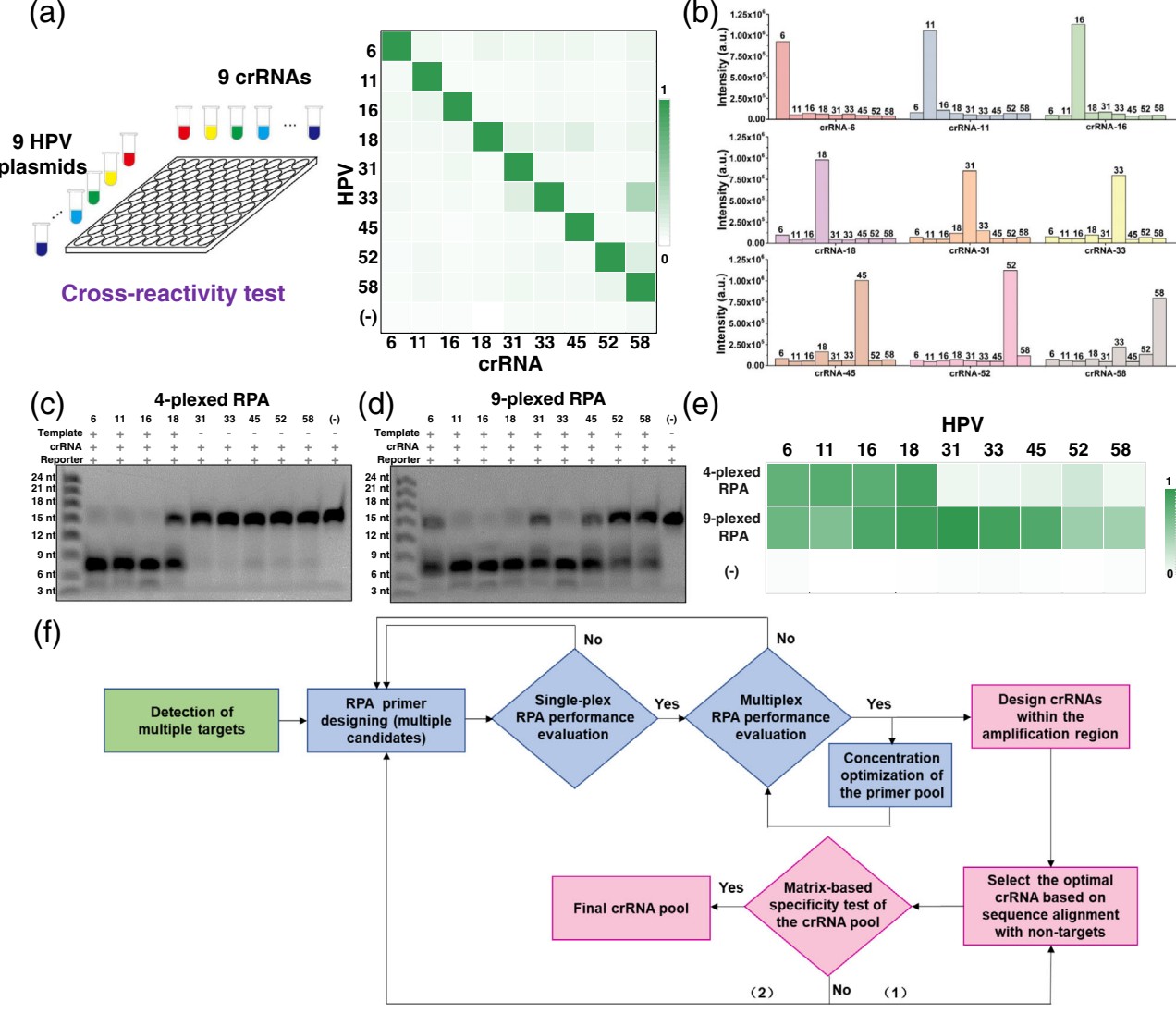

**Fig. 3 | Comprehensive characterization of the performance of the crRNA pool and multiplex RPA. a** Matrix-based reactivity test of the nine crRNAs against the nine HPV subtypes. The fluorescence-based test results were demonstrated as a heatmap. Source data are provided as a Source Data file. **b** Quantitative analysis of the reaction of individual crRNAs with the nine HPV subtypes. Source data are provided as a Source Data file. **c, d** Denaturing PAGE image showing the cleavage of the nine crRNAs recognizing the products of 4-plexed (**c**) and 9-plexed (**d**) RPA. Each experiment was repeated at least two times independently. **e** Heatmap showing the fluorescence-based characterization of the 4- and 9-plexed RPA products. To be noted, the amplicons used in c, d and e were from different batches of RPA assays. Source data are provided as a Source Data file. **f** Proposed procedure for RPA primer and Cas12-crRNA design for multiplexed target detection.

assay could be performed on the SS-Chip. The results (Fig. 5d) revealed similar fluorescence intensities in the 30 wells, further demonstrating the unbiased solution distribution and great mixing achieved by the device. These results laid a firm foundation for subsequent on-chip CRISPR/Cas detection assays.

To determine the optimum assay duration for the on-chip assay, the fluorescence intensities were collected at different time points. Positive signals (test sample) increased in a time-dependent manner, while those of the Control did not change significantly (Fig. 5e). In consideration of both the assay duration and signal intensity, $t = 15$ min was chosen as the readout time for the on-chip assay.

To evaluate the detection sensitivity of MiCaR, a series of unamplified and amplified HPV-16 plasmids were tested. A significant difference was detected for 0.1 nM of the unamplified plasmid versus the buffer (Fig. 5f). For the RPA-amplified HPV-16 plasmid, MiCaR could differentiate the blank sample from a $1 \times 10^{-18}$ M plasmid sample (Fig. 5g). As the fluorescence signals increased proportionally with the logarithm of plasmid concentrations ranging from 0 to $1 \times 10^{-16}$ M, the

limit of detection (LOD) was calculated to be $2.67 \times 10^{-19}$ M (0.26 aM) based on the 3σ/slope, where σ represents the standard deviation of readouts for three blank samples[33].

## HPV subtyping of patient samples using the MiCaR system

We next explored whether MiCaR can be used to analyze patient samples for HPV infection. To this end, 9-plexed HPV subtyping was performed using 100 human cervical swabs that were previously analyzed in a clinical laboratory (Supplementary Table 8). In the previous analysis, a color-coding-based PCR method was used for HPV subtyping. In this method, multiple color beads were used to differentiate between HPV subtypes, and the time taken was 3 h. In contrast, MiCaR used the space-coding approach to accurately identify multiple HPV subtypes with only one fluorophore (Fig. 6a). The Cas12a-crRNA mix was pre-loaded into the 30 designated wells, and the corresponding subtypes in the sample were recognized after RPA. The assay time required for MiCaR was about 40 min, including RPA, the on-chip reaction, and readout.

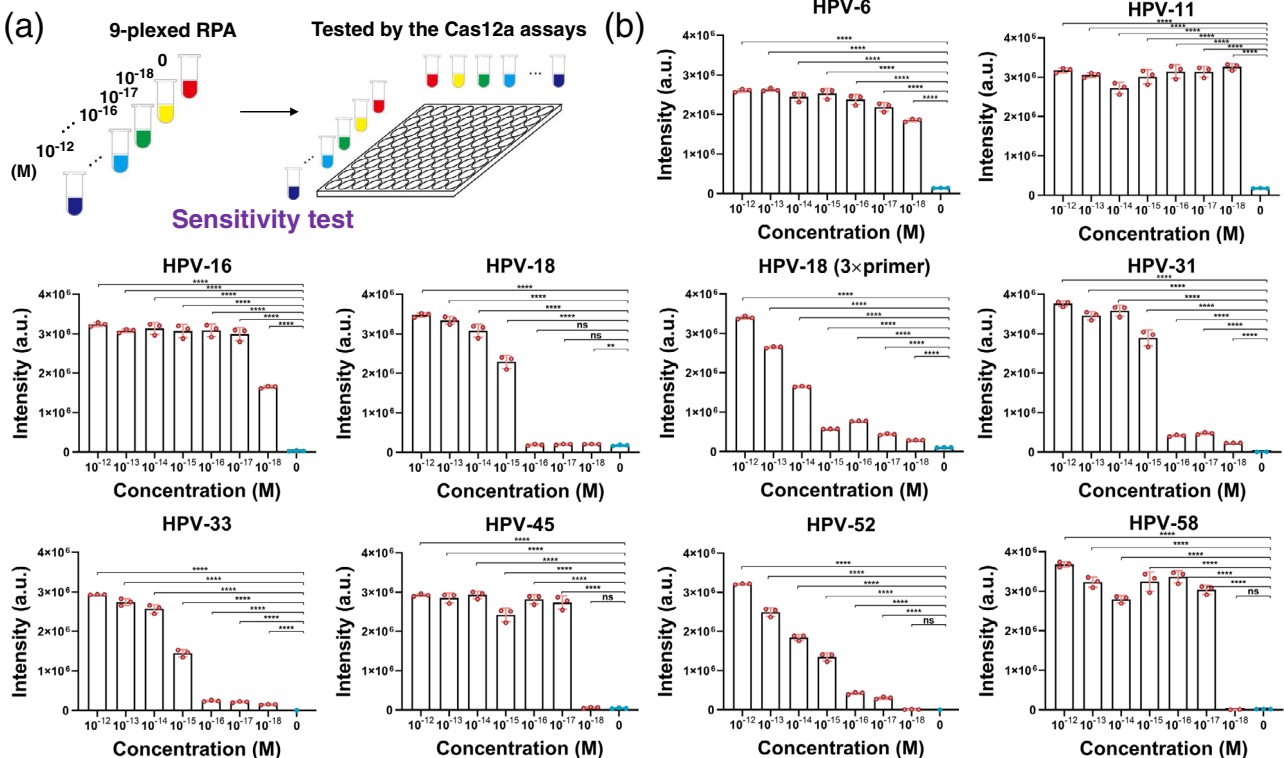

**Fig. 4 | Sensitivity test of the 9-plexed RPA assay for the nine HPV subtypes. a** Scheme showing that a series of samples with different plasmid concentrations of the nine targets were prepared, amplified and tested. **b** Titration of the plasmids after RPA based on the Cas12a assay. Note that the 9-plexed RPA assay (1× primer for each target) achieved a sensitivity of $10^{-17}$–$10^{-18}$ M for all the targets except HPV-18, while the sensitivity for HPV-18 could also be improved to $10^{-18}$ M after using 3×

HPV-18 primer in the 9-plexed assay. Values represent the mean ± SD of three independent experiments. Unpaired two-tailed $t$-tests were used to show the statistic difference between cohorts. The exact $P$-values were provided in Source Data. *$P$-value < 0.05, **$P$-value < 0.01, ***$P$-value < 0.001, and ****$P$-value < 0.0001. Source data are provided as a Source Data file.

Figure 6b shows the on-chip readout for a patient sample (#53). The wells (three as a group) designated for HPV-6, -16, and -45 showed significantly higher fluorescence than the other wells, suggesting that the sample was positive for these subtypes. The MiCaR-based quantitative results were compared to the previous clinical results for sample #53, demonstrating the high consistency between the two detection methods (Fig. 6c). The actual values of fluorescence intensity differed because the two approaches used different primers and detection mechanisms. The on-chip imaging readouts for several other samples are shown in Supplementary Fig. 8a. The 9-plexed HPV subtyping results of all 100 samples are shown in Fig. 6d, along with the clinical heatmap results (47 positive, 53 negative) (top) and the original on-chip imaging results (bottom). This comparison demonstrated the high concordance between the results of the two methods. The only exceptions were Samples #38 and #77. For Sample #38, the clinical results revealed the presence of HPV-16 and -52. However, according to MiCaR, the sample was positive for HPV-11 and -52. This difference is interesting, especially because one of the two recognized subtypes was common to the two assays. Nevertheless, we counted this as a false negative result based on HPV-16 discrimination. Moreover, MiCaR showed that Sample #77 was HPV-16-positive, while the clinical assay showed that this sample was negative for HPV. This false-positive could have resulted from contamination during sample transfer or detection.

To systematically analyze the performance of the MiCaR detection platform, a statistical analysis was conducted by comparing the on-chip results with those of the clinical assay. As shown in Supplementary Fig. 8b, the positive and negative results of all the nine subtypes were summed up in the pool of 100 samples, showing that their frequencies differed. Overall, MiCaR demonstrated a good performance in testing these patient samples, with a 97.8% positive

predictive agreement and 98.1% negative predictive agreement (Fig. 6e)[3,32]. The sensitivity and specificity were calculated to be 97.8% and 98.1%, respectively.

After testing clinical samples under laboratory conditions, we examined whether MiCaR could be used in the field, where access to proper heating incubators is limited. A steam eye mask, which is available even at home, was used to heat the device with a proper temperature (Supplementary Fig. 9a, b). The device was loaded with the CRISPR/Cas Reaction Mixes and a patient sample (#53). After 15 min of incubation, fluorescent images of the 30 outlet wells were obtained and analyzed. As shown in Supplementary Fig. 9c, the results obtained through this field-based assay with an alternative heat source were highly consistent with those produced using a normal heater (Fig. 6b, c). These findings revealed that MiCaR holds great potential in the point-of-care testing of multiple nucleic acids.

**Testing the generalizability of MiCaR with a respiratory virus panel (RVP)**

Lastly, we applied the MiCaR-based approach to the detection of eight most clinically relevant respiratory viruses[30,34]. This RVP includes influenza B virus (FLUBV), human coronavirus NL63 (HCoV-NL63), human coronavirus OC43 (HCoV-OC43), human respiratory syncytial virus (HRSV), human Coronavirus HKU1 (HCoV-HKU1), SARS-CoV-2, human parainfluenza virus serotype 3 (HPIV-3) and human metapneumovirus (HMPV). We first designed multiple pairs of RPA primers for each of the eight viruses (as shown in Supplementary Table 9), and the optimum primer was selected out based on the agarose gel electrophoresis (Supplementary Fig. 10, as indicated by the red numbers). Next, the crRNAs for recognizing the eight respiratory viruses were designed against the amplification region determined by the optimum

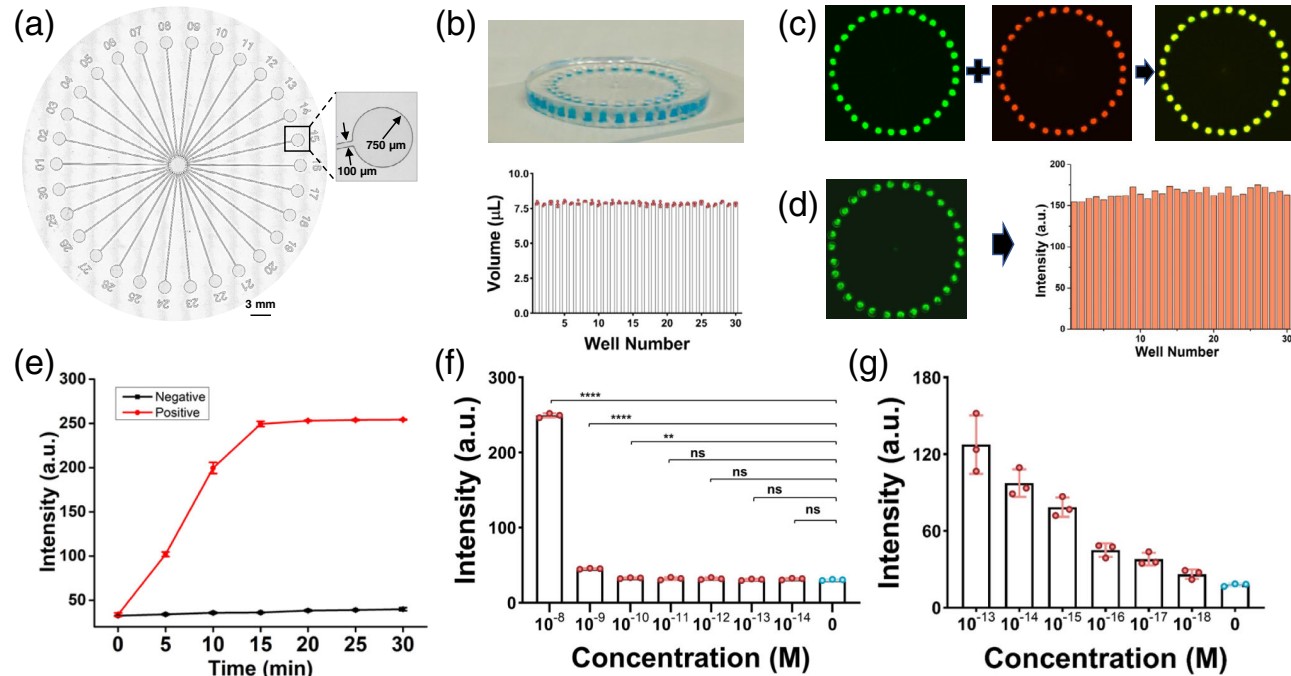

**Fig. 5 | Evaluation of SS-Chip performance. a** Stitched microscopic image showing the microchannels and wells of the SS-Chip, with labels. **b** Photograph of the device filled with blue food dye to mimic the sampling (top), and the corresponding quantitative analysis of the volume in each outlet (bottom). **c** Stitched fluorescence images showing the solution mixing performance of the SS-Chip with fluorescein (green) and sulforhodamine B (red). **d** Stitched green fluorescence image and the corresponding intensity analysis of Cas12a-based HPV-16 testing. Source data are provided as a Source Data file. **e** Kinetics of SS-Chip-based

detection of synthetic HPV-16 plasmids (positive, 10 nM; negative, 0). **f** Fluorescence intensities corresponding to the on-chip assay performed with increasing concentrations of the HPV-16 plasmid. **g** Analysis of the fluorescence signals of amplified HPV-16 plasmids. In **b** and **e**–**g**, values represent the mean ± SD and n = 3 biologically independent experiments. In **f** and **g**, unpaired two-tailed *t*-tests were used to show the statistic difference between cohorts. The exact *P*-values were provided in Source Data. **P*-value < 0.05, ***P*-value < 0.01, ****P*-value < 0.001, and *****P*-value < 0.0001. Source data are provided as a Source Data file.

RPA primers (Supplementary Table 10). The optimum crRNA with the proper secondary structure was selected out based on the software predication results. Then the performance of these crRNAs was evaluated by the 8 × 8 matrix-based activity test. Figure 7a shows that these crRNAs have great specificity against the relevant target. Subsequently, an 8-plexed RPA was performed and the products were verified by using the Cas12a assay. The results in Fig. 7b demonstrates that all the target viruses were successfully amplified and recognized. Furthermore, MiCaR was used to verify the 8-plexed RPA amplicons. The original detection images and bar plots are shown in Fig. 7c. All these results further proved that our approach could be used as a versatile strategy for multiplexed NAT.

## Discussion

The quick and cost-effective detection of multiple nucleic acids is essential for the efficient diagnosis of infectious diseases[16,28,29]. The current PCR-based gold-standard tests can only detect few targets. Moreover, they require a long turnaround time and complex instruments, which are usually available only in central laboratories[3]. Therefore, this method is not ideal for rapid and inexpensive NAT. Here, we report a unique assay called MiCaR that couples multiplex RPA with the CRISPR-Cas12a system and a 30-plexed SS-Chip to enable the rapid and low-cost identification of dozens of nucleic acid targets.

RPA can amplify nucleic acids efficiently at isothermal temperatures, greatly simplifying testing and reducing the requirement for expensive equipment[28]. Multiplex RPA has been widely used for NAT in previous study[19,26]. However, to our knowledge, the number of targets analyzed simultaneously in a single assay has been limited to a maximum of 4[1,16,27]. This limitation of RPA-based NAT could be due to two factors: primer design and target discrimination. According to the TwistDx RPA assay manual, there are no designing rules that guarantee

the good amplification performance of RPA primers (unlike PCR primers, which typically work well). The recommendations suggest that a series of candidate primer pairs should be designed and screened, and the optimal one should be selected. Additionally, it is possible that primers optimized for single assays will not perform well in multiplexed assays. Therefore, significant efforts are required to design and optimize RPA primers, especially when there are multiple targets. For example, a recent study aimed to develop an RPA primer pool for 13 HPV subtypes. However, only three of the 13 primers produced strong signals, and the others provided very weak amplification[35]. Ultimately, the researchers chose to modify commonly used PCR primer sets to perform the RPA assay and finally amplified and detected each 13 HPV subtype individually. In our study, to determine the optimal RPA primer combination for the nine HPV subtypes targeted by 9vHPV, several rounds of experiments, including primer design, sequence alignment, primer evaluation, and combination testing with the CRISPR/Cas12a assay, were performed (Fig. 2, Supplementary Figs. 1–3, Supplementary Data 1–2). The final primers showed great performance in single-, 4-, and 9-plexed RPA, with great efficiency and specificity, as indicated by the results of agarose gel electrophoresis, denaturing PAGE, NGS, and fluorescence measurement (Figs. 2, 3).

The other issue that needs to be addressed for multiplex RPA-based NAT is the accurate and simultaneous identification of signals for each target. The amplified targets can be identified individually, but this prolongs the assay duration and increases detection costs[32,35]. Most multiplexed amplification assays discriminate among targets using different fluorescent probes that contain regions complementary to the target amplicons[36,37]. Although a few fluorophores are available for NAT, the inherent spectrum overlap between them poses a barrier to sufficient multiplexed detection. Moreover, target identification based only on sequence paring can compromise

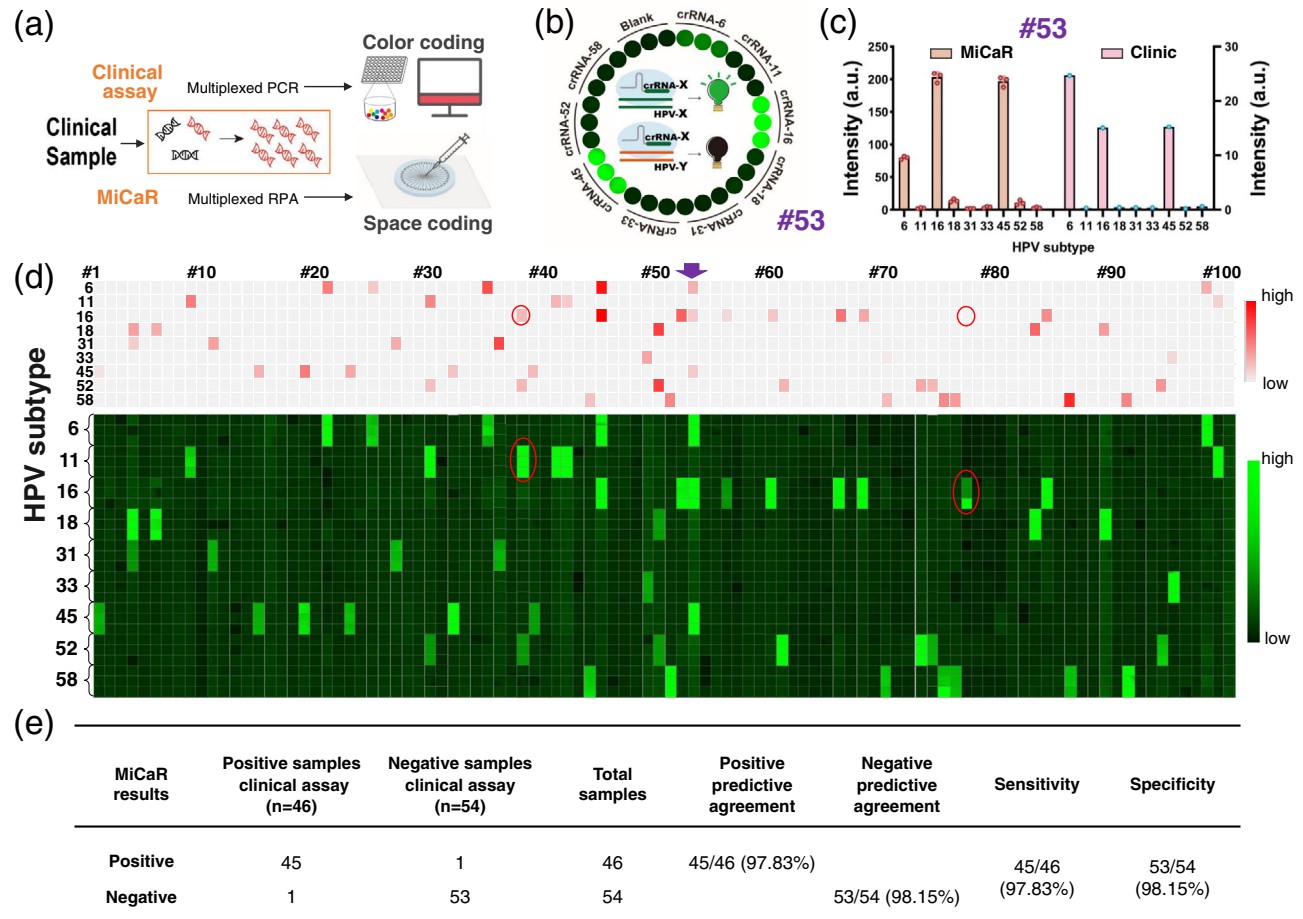

**Fig. 6 | Testing results of 100 patient samples for HPV infection. a** Brief schematic showing the differences between MiCaR and the clinical assay. The clinical laboratory conducting HPV subtyping assays using multiplexed PCR for target amplification and color coding for detection readouts. Meanwhile, in MiCaR, multiplex RPA was used for amplification, and microchip-based space coding was used for detection readouts. **b** Testing results of Sample #53, which was triple-positive for HPV subtypes, presented as a circle showing the original fluorescence images of the 30 outlet wells. **c** Quantitative analysis of results for Sample #53, tested using MiCaR and the clinical assay. Values represent the mean ± SD and $n = 3$

biologically independent experiments. Source data are provided as a Source Data file. **d** Parallel heat maps showing the results of the 100 samples based on the clinical assay and MiCaR-based testing. Each sample was tested in triplicate using MiCaR. The purple arrow indicates Sample #53. The red circles indicate the inconsistencies (Sample #38 and #77) between MiCaR and the clinical assay. Source data are provided as a Source Data file. **e** Positive predictive agreement (PPA), negative predictive agreement (NPA), sensitivity and specificity of MiCaR for the detection of the nine HPV subtypes in clinical samples.

detection accuracy owing to off-target amplification[7]. Herein, we coupled the CRISPR-Cas12a system with microfluidics technology to conquer these challenges. High specificity was achieved through Cas12a–crRNA binding and recognition[38]. We screened multiple available crRNAs across the L1 gene of all nine HPV subtypes according to basic designing principles. During the first round of crRNA testing, cross reactivity between HPV-6 crRNA-I and the HPV-11 plasmid was detected (Supplementary Fig. 5, Supplementary Table 6). Subsequently, comprehensive comparisons were performed using these potential crRNA sequences and other non-target L1 genes to ensure a difference of at least three nucleotides, which could prevent the crosstalk between a crRNA and the non-target subtypes. After this round of checks, the best crRNAs against the nine targets were selected. Finally, we tested and confirmed the specificity of each crRNA by designing a matrix test of 9 crRNAs × 9 targets and performed 4- and 9-plexed assays (Fig. 3a–e). The multiplexed assays also demonstrated a great sensitivity of ~attomole for the nine HPV subtypes (Fig. 4). Given the significant effort to obtain an optimal RPA primer pool and coupled Cas12-crRNA panel for detecting multiple targets, as mentioned previously, we proposed a procedure for primer designing in such cases. The steps are summarized in Fig. 3f and could be used in a

variety of such studies. For example, we also successfully applied this strategy to testing the eight common respiratory viruses (Fig. 7a, b).

Microfluidics technology has been extensively used for molecular detection[39–42]. However, common microfluidic platforms have complicated fluidic networks, and often require valves, accessory pumps or other complex controlling systems[43,44]. Thus, they are not ideal for the quick and on-site detection of multiple targets[45]. For example, some droplet- and electrochemistry-based microfluidic biosensors were proposed for nucleic acid detection, but the targets that can be analyzed simultaneously are limited[46,47]. Though CARMEN and mCARMEN enables the testing of over one hundred nucleic acid targets, the relatively complex setup or expensive instrumentation hinders their wide use. To aid the identification of multiple targets, we developed a simple 30-plexed microfluidic SS-Chip. This chip has an undecorated hub-spoke network that can efficiently divide the sample into smaller, equal volumes after simple pipetting. The amplified sample is allowed to react with a pre-loaded Cas12a/crRNA Mix inside space-coded wells, allowing the precise identification of multiple targets with only one fluorescent probe. The SS-Chip can simultaneously provide three readouts (as a group) for each of the nine targets, thus having a significant advantage with respect to assessing detection performance.

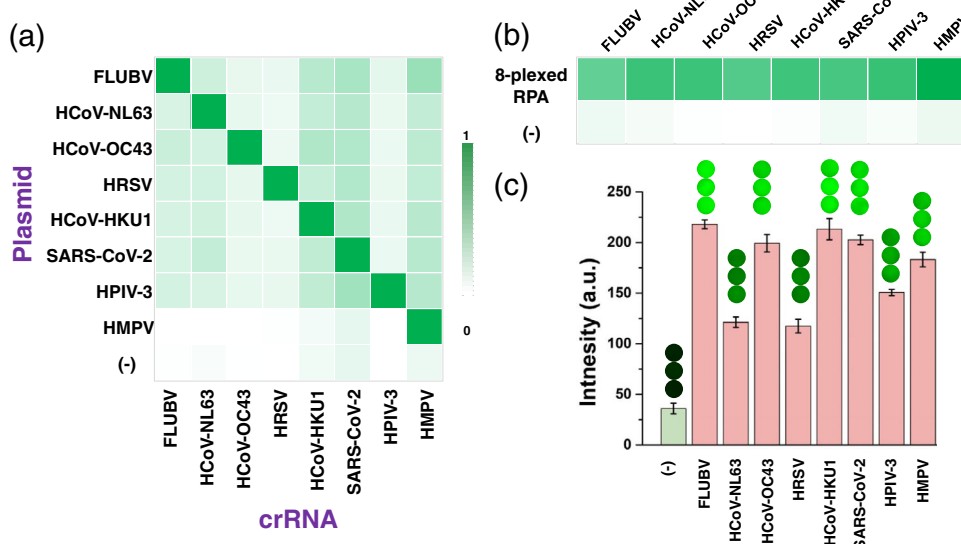

**Fig. 7 | Testing of the RVP with the MiCaR-based approach. a** Discrimination of the eight respiratory viruses using the Cas12a-based assay. (−) was used as Control, in which no plasmid was added. Source data are provided as a Source Data file. **b** Evaluation of the 8-plexed RPA products based on the Cas12a-based assay. The template was $10^{-12}$ M plasmid for each virus. (−) was used as Control, in which no plasmid was added during the RPA assay. Source data are provided as a Source Data file. **c** MiCaR-based readout to evaluate the 8-plexed RPA assay. (−) was used as Control, in which no crRNA was added. To be noted, the amplicons used in **b** and **c** were from different batches of RPA assays. Values represent the mean ± SD and $n = 3$ biologically independent experiments. Source data are provided as a Source Data file.

The on-chip assay takes only 40 min and its LOD reaches $2.67 \times 10^{-19}$ M, which are significantly superior to RT-PCR (>2 h, LOD ~$1.6 \times 10^{-18}$ M)[28]. In this study, we tested 100 patient samples, and the on-chip results were highly consistent with clinic results, showing a specificity of 97.8% and sensitivity of 98.1%. The differences in fluorescence for three independent measurements in the same group were minimal, indicating the accuracy and stability of MiCaR-based detection (Figs. 6, 7c). Further, we demonstrated the potential of MiCaR as a point-of-care tool for applications in the field, wherein access to sophisticated instruments is limited.

The global outbreak of COVID-19 highlights the critical need for the fast and reliable detection of multiple pathogens (e.g., to discriminate between respiratory viruses and mutated strains) in the laboratory and clinic and even at home[25,48]. Therefore, the efficient identification of target pathogens can facilitate the management and containment of relevant infectious diseases[48–50]. Though we only demonstrated the proof-of-concept by testing the nine HPV subtypes and eight respiratory viruses, the 30-plexed SS-Chip can detect 30 targets simultaneously. The space-coding based strategy is straightforward and displays results using only one fluorescent probe. It is also very convenient to expand the multiplexing capability of this chip by directly adding more microchannels and wells. Another direction to further improve the MiCaR system is to optimize the amplification and detection conditions and generate a one-pot assay, which should further ease the operation and reduce the duration time. Moreover, automatic operation could be made possible by integrating an undecorated microelectromechanical setup with the MiCaR system[51], as the isothermal amplification and single-probe based systems can simply be mounted.

In summary, we developed a multiplexed NAT system, named MiCaR, by leveraging the great performance of multiplex RPA for amplification, the high specificity of CRISPR/Cas12a system in target recognition, and the appreciable utility of a microfluidic device for liquid handling. Using a simple procedure and short assay duration, MiCaR can detect multiple targets. MiCaR also showed reliable and accurate detection results when compared with clinical reports. Hence, we anticipate that MiCaR will be used widely for rapid and sensitive NAT across a range of biotechnology and healthcare-related applications.

## Methods
### Ethical statement
The study was approved by the Committee on Human Research of Maternal and Child Health Hospital of Hubei Province (2020IECXM045) and China's Ministry of Science and Technology. All research was performed in accordance with relevant guidelines and regulations. Informed consent was obtained from all participants.

### Materials
A TwistAmp Basic kit containing RPA freeze-dried pellets was purchased from TwistDx (Cambridge, UK). *Lachnospiraceae bacterium* ND2006 Cas12a (LbCas12a) was purchased from Meige company (Guangzhou, China). Plasmids containing nine types of the HPV L1 gene, relevant gene sequences of the eight respiratory viruses and TBA11-FQ were obtained from Tsingke Company (Beijing, China). The primers and crRNAs used were synthesized by Sangon Biotech Company (Shanghai, China). DNA agarose and nucleic acid gel stains were purchased from Yeasen Company (Shanghai, China). Fluorescein and sulforhodamine B were purchased from Shanghai Aladdin Biochemical Technology Co. Ltd. (Shanghai, China). The steam eye mask was purchased from Kao Commercial (Shanghai, China).

### RPA primer and CRISPR crRNA designing for the HPV panel
Oligonucleotides were designed, and structural prediction and ΔG calculation were performed using the mFold webserver (http://www.unafold.org/mfold/software/download-mfold.php) and OligoAnalyser (https://eu.idtdna.com/calc/analyzer). The RPA primers for the nine subtypes of HPV were designed following the instructions of the TwistAmp Assay Design Manual (https://www.twistdx.co.uk/wp-content/uploads/2021/04/twistamp-assay-design-manual-v2-5.pdf) using the DNAMAN 9 (https://www.lynnon.com/) and CE Design 1.03 (http://www.ce-mark.com/ce-design.htm) tools. The crRNA candidates for the nine types of HPV were designed using the common principles of Cas12a crRNA design (https://international.neb.com/faqs/2018/05/

03/how-do-i-design-a-guide-rna-for-use-with-engen-lba-cas12a). These HPV L1 genes were checked and found to be conserved using NCBI data through multiple sequence alignments analysis.

## Single-plex RPA assay

RPA reactions were carried out using a commercially available kit (https://www.twistdx.co.uk/wp-content/uploads/2021/04/INTABAS-v3.0-TwistAmp-Basic-Kit-Quick-Guide-INTABAS-1-1.pdf) according to the manufacturer's instructions. One freeze-dried RPA pellet provided the RPA mix required for one reaction. First, the rehydration solution was prepared as follows: forward primer (2.4 µL, 10 µM), reverse primer (2.4 µL, 10 µM), rehydration buffer (29.5 µl), DNA template (5 µL, the concentration is changed for different purposes), and ddH$_2$O (8.2 µL). This mixture was vortexed and then spun briefly. The rehydration solution was transferred to the reaction pellet and mixed using a pipette until the entire pellet had been resuspended. Then, magnesium acetate (2.5 µL, 280 µM) was added, and the solution was mixed well. The amplification reaction was performed at 39 °C for 20 min. For the negative control, the reaction was set up using ddH$_2$O instead of the DNA template. Thereafter, the RPA products were characterized using different methods or stored at −20 °C for subsequent Cas12a-based cleavage assays.

## Optimization of primer pool concentration

Both the forward and reverse primers used were 2.4 µL (10 µM), resulting in a total primer concentration of -1.0 µM in the 50-µL single-plex RPA assay. The instructions of the RPA kit mentioned that the primer constitution must be optimized during multiplexed amplification to avoid primer aggregation and obtain sufficient amplicons for each target. The primer pool was prepared by adding equimolar quantities of the forward and reverse primers for all nine HPV subtypes. First, the forward and reverse primer pool for each subtype was prepared (1 µM each), resulting in a total primer concentration of 9 µM in the assay. The results showed that the RPA assay for the nine targets did not work due to the formation of primer dimers. Hence, the primer concentration needed to be reduced. Subsequently, forward and reverse primer pools with three different total concentrations of 0.1, 1.0, and 5 µM were prepared for testing amplification performance (using the HPV-16 plasmid as the target to find the optimal pool). The primer pool, which contained all nine types of forward and reverse primers (2.4 µL; 1, 10, or 50 µM), rehydration buffer (29.5 × 9 µl), and ddH$_2$O (8.2 × 9 µL), was prepared once for nine reactions in one centrifuge tube. Then, it was divided into nine equal parts for later use.

## 4- and 9-plexed RPA assay

First, all the nine types of forward and reverse primers (2.4 µL; 10 µM), rehydration buffer (29.5 × 9 µl), and ddH$_2$O (8.2 × 9 µL) were mixed. The solution was divided into nine equal parts. Each part was then vortexed and spun briefly. The rehydration solution was transferred to the reaction pellet and mixed using a pipette until the entire pellet had been resuspended. For the 4-plexed RPA, the DNA templates of HPV−6, HPV-11, HPV-16, and HPV-18 were added. The concentration of each subtype was $1.0 \times 10^{-11}$ M. For the 9-plexed RPA, the DNA templates of all nine HPV subtypes were added, with the concentration of each being $1.0 \times 10^{-11}$ M. Then, magnesium acetate (2.5 µL, 280 µM) was added and the solution was mixed well. The amplification reaction was performed at 39 °C for 20 min.

## Cleavage assay in a centrifuge tube or 96-well microplate

The cleavage assay was performed based on previous studies[19,32]. The assay buffer contained 10 mM Tris, 70 mM KCl, and 10 mM MgCl$_2$ and had a pH of 7.9. Before the assay, the TBA11-FQ reporter oligonucleotides were heated at 95 °C for 10 min and cooled down before use. The assay procedure was as follows. First, LbCas12a (2 µL, 2 µM), crRNA (4 µL, 1 µM), and the cleavage buffer (29 µL) were preincubated at 37 °C

for 10 min to generate the LbCas12a/crRNA complex. Then, the target DNA (5 µL, 10 nM plasmid or RPA product) or cleavage buffer (5 µL, as negative control) and the TBA11 reporter (10 µL, 1 µM) were added to the previous solution to generate a 50-µL reaction system. The cleavage reaction was performed at 37 °C for 15 min. Subsequently, the cleavage solution was heated at 65 °C for 10 min to stop the reaction.

## Fluorescence measurement of cleavage assay results

The cleavage results were characterized on a microplate reader (SpectraMax i3x, Molecular Devices, CA, USA). The excitation wavelength was set to 488 nm, and fluorescence emission was collected at 518 nm.

For each HPV subtype, multiple crRNA candidates could be designed based on the L1-gene according to the basic designing principle of Cas12a-crRNA. However, the designed crRNA for a specific subtype could recognize other subtypes due to the high homology among different HPV subtype sequences. Therefore, experiments were conducted to screen for crRNA cross-reactivity. A careful comparison of the crRNA with the sequences of non-target HPV subtypes was performed to identify crRNAs that could differentiate between the target and non-targets. The cross-reactivity of the nine crRNAs was tested against the L1 gene plasmid of the nine HPV subtypes using a 9×10 cleavage assay. This assay also included a group of controls without any plasmid. The cleavage assay solutions were prepared in a 96-well plate following the protocol described above. The fluorescence obtained was measured using a microplate reader.

## Sensitivity test of the 9-plexed RPA assay for the HPV panel

Plasmid samples with different concentrations ($10^{-12}$, $10^{-13}$, $10^{-14}$, $10^{-15}$, $10^{-16}$, $10^{-17}$, $10^{-18}$ and 0 M) were prepared as the templates (each sample was a mixture of the nine HPV subtypes with the relevant concentration). Then a multiplex RPA assay (forward and reverse primer: 0.11 µM for each target, 2.4 µL of total volume) were carried out for each sample. The RPA products were evaluated with the Cas12a assay. For HPV-18, we also increased its primer to 3 folds of the primary concentration and performed the 9-plexed RPA assay.

## Denaturing PAGE

LbCas12a (2 µL, 2 µM) and crRNA (4 µL, 1 µM) were preincubated in a total of 20 µL buffer (70 mM K$^+$) at 37 °C for 10 min. Then FAM-labeled TBA11 (25 µL, 10 µM) and the product of the 4- and 9-plexed RPA (5 µL, 50 nM) were added into the reaction solution and the mixtures were incubated at 37 °C for 30 min. The system was heated at 65 °C for 10 min to stop the cleavage reaction. Then the solution was mixed with the loading buffer and loaded into a 20% denaturing PAGE gel containing 8 M urea. Electrophoresis was carried out at 120 V (about 40 V/cm) for 90 min (Mini-PROTEAN Tetra Cell system, Bio-Rad) in the 1×TBE buffer. The gel was scanned using Bio-rad ChemiDoc MP (170−8280) (BioRad Company, Shanghai, China) to obtain the readout.

## Agarose gel electrophoresis

Agarose gel electrophoresis was performed to analyze the products of RPA. The RPA products were separated with agarose gel (3.5%, w/v, pre-stained with GelRed) at 150 V/180 mA for 1 h. Finally, the DNA agarose gel was exposed to UV (BioRad Company, Shanghai, China) to take the gel graph.

## Next-generation sequencing

Though agarose gel can roughly identify the size of the RPA products, it cannot provide detailed information about the sequence of the multiplex RPA products. To characterize the sequence of the amplicons of the nine HPV subtypes, the products of the multiplex RPA were sent to Sangon Biotech company and identified by next-generation sequencing (NGS).

## Fabrication of the Starburst-Shaped Chip (SS-Chip)

The SS-Chip was fabricated using standard soft-lithography technique[52,53]. First, negative photoresist SU-8 3025 was spin coated on a silicon wafer (800 rpm, 40 s; 3000 rpm, 60 s; corresponding to a height of ~25 μm), followed by a prebake (65 °C, 10 min; 95 °C, 25 min). After the UV-exposure (8 s, 5 mJ/cm$^2$), the photoresist was post-baked (95 °C, 5 min) and developed. A hard bake (135 °C, 60 min) was processed to obtain the final mold. A mixture consisting of PDMS and curing agent (10:1) was poured onto the mold. After a heating process (65 °C, 2 h), the PDMS slab was cut and punched to get the inlet and outlets. Then a glass slide (75 mm × 50 mm) was treated with Aquapel (Aquapel Glass Treatment, PA, USA) to make it hydrophobic (see the following for more details), and served as the substrate. The prepared PDMS was attached (reversible bonding) onto the hydrophobic glass slide without plasma treatment. The PDMS could be detached and reused for >20 times after thorough alcohol wash and nitrogen blow drying.

## Aquapel treatment

To perform the on-chip assay, we first load different Detection Master Mixes (DMMs, including Cas12a, different crRNAs, and the reporter) in the relevant outlet wells, and then inject the sample from the central inlet to disperse it into each outlet. To avoid the DMM solution to flow back from the outlet into the inlet and cause unwanted cross contamination during the testing, Aquapel was used to make the microchannels hydrophobic. This ensures the DMM solution to stay in the designated outlet and facilitates the mixing of DMM and the sample.

It is simple to carry out the Aquapel treatment. The treatment procedure is: use a pipette to drop ~1 mL Aquapel onto the surface of the glass slide, spread the Aquapel evenly with the pipette tip, wipe out the extra Aquapel on the glass slide after 10-min drying.

## Characterization of the SS-Chip

To evaluate the SS-Chip's ability of solution aliquoting, green food dye (240 μL) was injected from the central inlet. The solution flowing into the 30 outlets was withdrawn, and the solution volume from each outlet was measured via a 10-μL Hamilton syringe. To mimic solution mixing during the cleavage reaction, fluorescein solution (10 μM, 4 μL) was pre-loaded into each well. Subsequently, sulforhodamine B solution (3 μM, 120 μL) was injected from the inlet. After 5 min, the fluorescence images from the green and red channel were captured for each outlet.

## On-chip cleavage assay

First, Cas12a (100 nM) and crRNA (100 nM) were pre-incubated at 37 °C for 10 min. Then, the TBA11-FQ reporter (2.5 μM) was added to prepare the Detection Master Mix (DMM). The total volume of the DMM depended on how many wells were to be used for testing. Typically, for each well, 5 μL DMM was added separately. To test in triplex, the mix was added to three adjacent wells. Then, 150 μL solution containing one or more HPV subtype plasmids (5 nM for each) was injected from the central inlet and divided within the 30 outlets and allowed to react with the pre-loaded DMM. Each well had 5 μL of the DMM and 5 μL of the target. The SS-Chip was incubated at 37 °C for a certain period (depending on the goal of the test; typically, the duration was 15 min). Subsequently, fluorescence images were obtained for each well.

## Limit of detection (LOD) of the SS-Chip

The detection sensitivity of the SS-Chip for plasmids without pre-amplification was first characterized. A series of HPV-16 plasmid solutions with concentration from 0 to 10 nM were tested. Five microliters of these samples were added to the outlet wells (each sample in triplicate). Then, 150 μL DMM for HPV-16 was injected from the central inlet, and the device was incubated at 37 °C for 15 min. Subsequently, imaging was performed.

Next, the LOD for testing plasmids after pre-amplification was examined. The template HPV-16 plasmid was diluted to $10^{-13}$ M, $10^{-14}$ M, $10^{-15}$ M, $10^{-16}$ M, $10^{-17}$ M and $10^{-18}$ M and amplified via RPA as described above. Five microliters of the RPA product of each sample were added to the outlet wells (each sample in triplicate). For the negative control, 5 μL ddH$_2$O was added instead of the RPA product. Then, 150 μL DMM for HPV-16 was injected from the central inlet, and the device was incubated at 37 °C for 15 min. Subsequently, imaging was performed.

## Preparation of clinical sample

Human cervical cell specimens ($n = 100$) used in this study were collected in multiple times from Maternal and Child Health Hospital (MCHH) of Hubei Province, Huazhong University of Science and Technology. The study was approved by the MCHH Committee on Human Research (2020IECXM045). These samples were screened in the clinical laboratory for HPV infection by PCR assays (Tellgen Corporation, Shanghai, China) prior to our assay. The collected samples were anonymized and there is no personal identification data for the individuals. The collected samples were heated at 95 °C for 10 min to release the HPV DNA and stored at −20 °C before use.

## Clinical sample testing based on multiplex PCR

The samples were screened in the clinical laboratory for HPV infection by multiplex PCR assays (Tellgen Corporation, Shanghai, China) prior to our assay. Briefly, the HPV DNA released from the sample was amplified by multiplex PCR with biotin-labeled primers. Then the amplicons were hybridized to color-coded microspheres coated with HPV subtype-specific probes. Next, the microspheres were incubated with phycoerythrin (PE)-conjugated streptavidin (SA-PE). After through wash, the microspheres were read on a Luminex 200 system (Luminex Corporation, Texas, USA). The HPV subtypes were determined based on the fluorescent dye signature carried by the microspheres.

## On-chip sample testing

First, the samples were amplified via multiplex RPA. The procedure was similar to that described above. One freeze-dried RPA pellet provided the RPA mix required for one sample. First, the rehydration solution was prepared: equimolar quantities of the nine forward primers for the nine subtypes (0.11 μM for each, 2.4 μL of total volume) were mixed with equimolar quantities of the corresponding nine reverse primers (0.11 μM for each, 2.4 μL of total volume), rehydration buffer (29.5 μl), sample (5 μL), and ddH$_2$O (8.2 μL). After through mixing, the rehydration solution was added to resuspend the reaction pellet. Following the addition of magnesium acetate (2.5 μL, 280 μM), the amplification reaction was performed at 39 °C for 20 min. For the negative control, the reaction was set up using ddH$_2$O instead of the sample.

Then, the amplified sample products were tested on the SS-Chip. DMMs for the nine HPV subtypes were prepared as described above in a 1.5 mL centrifuge tube. For the negative control, crRNA was replaced with buffer. The tubes with the DMMs were incubated at 37 °C for 10 min and then transferred to an ice box. Then, 5 μL of the DMMs were added to the 30 reaction wells, with HPV-6, HPV-11, HPV-16, HPV-18, HPV-31, HPV-33, HPV−45, HPV-52, HPV-58, and Negative Control loaded into wells #1–#3, #4–#6, #7–#9, #10–#12, #13–#15, #16–#18, #19–#21, #22–#24, #25–#27, and #28–#30, respectively. The multiplex RPA product of each sample (50 μL) was diluted to 150 μL and injected from the central inlet and mixed with the DMM at each outlet. Then, the SS-Chip was incubated at 37 °C for 15 min and imaged to obtain the readout.

## Optical imaging system and image process

EVOSTM M5000 imaging system was used to take the fluorescence images. Each reaction well was imaged under the 10× microscope and a

total of 30 pictures were obtained. The acquired pictures were further analyzed using Image J 1.8.0 and Origin 9. To quantitatively measure the fluorescence value of each reaction well, the image was loaded into the Image J software. Then, a few steps were processed as: "Image", "Color", "Split Channels", "Choosing Green channel" and "Analysis of the value of green".

### Design and evaluation of the respiratory virus panel (RVP)

Eight of the most clinically relevant viruses, including influenza B virus (FLUBV), human coronavirus NL63 (HCoV-NL63), human coronavirus OC43 (HCoV-OC43), human respiratory syncytial virus (HRSV), human Coronavirus HKU1 (HCoV-HKU1), SARS-CoV-2, human parainfluenza virus serotype 3 (HPIV-3) and human metapneumovirus (HMPV), were selected. The RPA primers for the RVP were designed following the instructions of the TwistAmp Assay Design Manual using the DNAMAN 9 and CE Design 1.03 tools, similar to the HPV panel. Multiple pairs of primers were designed to target specific gene of each virus (FLUBV, PB1 gene; HCoV-NL63, Orf1ab gene; HCoV-OC43, Orf1ab gene; HRSV, M gene; HCoV-HKU1, Orf1ab gene; SARS-CoV-2, Orf1ab gene; HPIV-3, M gene; and HMPV, F gene; the gene sequences are listed in Supplementary Data 5). Then, single-plex RPA assay was carried out for each pair of primers with a plasmid template ($10^{-10}$ M, 5 μL). The RPA products were characterized using agarose gel electrophoresis. The optimum primer that produced a relatively denser band was selected out.

Then the crRNA candidates for recognizing the eight respiratory viruses were designed against the amplification region determined by the optimum RPA primers. Next, structural prediction and ΔG calculation were performed using the mFold webserver and OligoAnalyser. The optimum crRNA with the proper secondary structure was selected out based on the predication results. Then the crRNAs were synthesized and evaluated by the 8 × 8 matrix-based activity test. Every crRNA was tested by using the eight respiratory viruses, and (−) was used as Control, in which no plasmid was added.

Next, the 8-plexed RPA assays were performed to amplify the plasmids with a concentration of $10^{-12}$ M for each respiratory virus. The amplicons were first tested by the Cas12a-based assay, with the results measured by using a microplate reader. (−) was used as Control, in which no plasmid was added during the RPA assay. Furthermore, MiCaR was applied to measure the products of the 8-plexed RPA assay. To test in triplex, the DMM was added to three adjacent wells. (−) was used as Control, in which no crRNA was added. Then, 150 μL solution containing the 8-plexed RPA products was injected from the central inlet and divided into the 30 outlets to react with the pre-loaded DMM. The SS-Chip was incubated at 37 °C for 15 min and the fluorescence images were captured for each well.

### Statistics and reproducibility

Unless specified otherwise, statistical analysis was performed using Origin 9. Standard deviations and mean values were calculated using data from at least three identical assays. The unpaired two-tailed $t$-test was performed in Excel (Microsoft Office 2019) to compare fluorescence signals of two cohorts. A $P$-value < 0.05 was considered to be statistically significant. Positive predictive agreement (PPA), negative predictive agreement (NPA), sensitivity, and specificity were calculated using MedCalc (https://www.medcalc.org/calc/diagnostic_test.php).

No statistical method was used to predetermine sample size. No data were excluded from the analyses. The experiments were not randomized. The investigators were not blinded to allocation during experiments and outcome assessment.

### Reporting summary

Further information on research design is available in the Nature Research Reporting Summary linked to this article.

## Data availability

The data supporting the results in this study are available in the Supplementary Information and Supplementary Data files. The raw fluorescence data for all tests are provided in the Source data file. Source data are provided with this paper.

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

## Acknowledgements

We gratefully acknowledge the financial supports from National Natural Science Foundation of China (22174150, Y.L.; 21904139, Y.L.; 22074152, Y.Y.; 21921004, Y.Y.; 21991080, M.L.), Natural Science Foundation of Hubei Province (2019CFB391, D.C.).

## Author contributions

Z.X. performed most the experiments and prepared the draft. D.C. collected the clinical samples, provided the clinical reports and assisted the sample testing. T.L. assisted the RPA/crRNA design and optimization. J.Y. assisted the on-chip sample testing. J.Z. assisted the gel-based experiments and revised the manuscript. T.H. and R.H. helped the device fabrication, modification and operation. Y.L. conceived, designed and managed the project. Z.X. and Y.L. analyzed the data and prepared the manuscript. Y.Y. and M.L. provided suggestions during the project and also provided funding. All authors contributed to the revision of this manuscript.

## Competing interests

There are no competing interests to declare.
