## [Peer Review File · Nature Communications]

REVIEWER COMMENTS

Reviewer #1 (Remarks to the Author):

In this manuscript, Xu et al proposed a new microfluidic platform that couples CRISPR/Cas12a and multiplex RPA for the testing of multiple nucleic acids. Considering current pandemic, it should be very meaningful to have such a simple but efficient tool for rapid diagnosis of infectious diseases. This strategy realizes the multiplexed detection relying on the "microfluidic-based space coding", which utilizes the specific spatial spots to indicate each individual targets after the recognition of relevant crRNA. The idea is smart as it requires only one probe instead of multiple ones efficiently avoiding the spectrum overlap problem. Another important impact of this manuscript is the protocol proposed for designing the RPA primer pool and the crRNA pool. The authors performed a comprehensive study to show how to obtain an appropriate RPA pool first, and then the crRNA pool, and at the end look back to coordinate the two pools to reach the final panel. It is nice to see such a logic strategy to reach a success for the detection of multiple nucleic acid targets based on an isothermal amplification method (i.e., RPA). The authors demonstrated that they realized the simultaneous amplification and detection of the 9 HPV subtypes in 40 min, which I think is very promising because such a detection way efficiently decreased the cost and the operation bias compared to the single-plex detection. Finally, the authors showed an impressive data for the detection of 100 patient samples with potential HPV infection risk. The triplicate test for the same HPV subtype ensured detection accuracy and reliability. This is useful in real applications to efficiently reduce false positive and false negative results. I think this study is of great impact to the field and also of great interest to Nature Communications' readers. I recommend its acceptance after addressing the following minor issues.

1. When describe the SS-Chip, "Inlet", "inlet", "Outlet" and "outlet" were mixedly used. Please check the whole manuscript and format the use.
2. Why use Aquapel treatment to make the device hydrophobic? This part is not clear. And how to carry out the treatment? Please give more details.
3. In discussion, it is recommended to add more details to compare the SS-Chip and other microfluidic platforms, which helps the readers further understand the unique advantages of this work.
4. It is interesting to demonstrate the usage of an eye mask for potential point-of-care testing. How to know that the steam mask can heat the SS-Chip at 40 °C for 20 min? It is recommended to find a way to measure the temperature of the mask and the device, and add data to support it.
5. In Figure 3d, e, f, is the heatmap correlated with the gel results? It seems that the intensities do not match between the two groups of results. Please clarify.
6. There are some typos (e.g., Page 26, the PDMS slab to was cut and....) in the manuscript. Please check.
7. The references are not in the same format.

Reviewer #2 (Remarks to the Author):

This manuscript by Xu, Chen, and coworkers describes a microfluidic approach for multiplexed nucleic acid detection using recombinase polymerase amplification (RPA) and Cas12a-based detection. The authors use their approach, MiCar, to develop an assay for the detection of 9 HPV subtypes. The authors then test their assay on a set of 100 patient samples, and demonstrate high concordance with a clinical multiplexed PCR assay.

Overall, I appreciate the work the authors have done to multiplex RPA and combine it with

Cas12a detection. However, additional work is needed to validate the performance of the HPV assay, as well as to demonstrate that the author's approach can be generalized to detect other pathogens.

Major comments:

1. What is the benefit of doing multiplexed RPA rather than multiplexed PCR? The authors justify this in the introduction by mentioning that RPA is fast and compatible with one-pot detection, but they do not (to my knowledge) show any one-pot data. The authors should include additional justification for the use of RPA in the introduction.

2. A key question is the generalizability of the author's approach for designing and optimizing primer sequences for multiplexed RPA. The authors should demonstrate that their approach can be applied to at least one other multiplexed panel.

3. How well does the multiplexed RPA amplification work? The authors appear to have demonstrated it using only a single target concentration of $1e-11$ M. This is a very high target concentration (>1 million copies per microliter). The authors need to perform more rigorous analysis of the sensitivity of their method for detecting each of the HPV subtypes on their panel to establish the limit of detection of their assay.

4. The background intensity (in arbitrary units) appears to vary widely between figure panels. For example, in Figure 4e and 4f the background is ~ 90 a.u., in Figure 4g the background is < 30 a.u., whereas in Figure 5xx the background is ~ 50 a.u.. Some of the samples from the dilution series in Figure 4g have fluorescence values in this range. The authors should more rigorously characterize the amount of background fluorescence in their assay to ensure that this does not influence assay performance.

Minor comments:

1. In the introduction, there are a few inaccurate statements (for example, CARMEN uses Alexa Fluor dyes, not FAM/HEX/TEX/Cy5). Also, the authors do not mention mCARMEN, which allows for microfluidic-based multiplexing using Cas13 and Cas12 without requiring a complex workflow or dye-based color coding (<https://www.nature.com/articles/s41591-022-01734-1>).

2. In Figure 2c, why is the amplicon fidelity low for HPV11 and HPV18?

3. Figure 3a is confusing, and should be redrawn to make the author's approach clearer.

4. I am confused by the results shown in Figure 4e - for some reason the "negative" sample appears to be increasing in fluorescence over time whereas the "positive" one remains flat.

5. Figure 6 appears to be a summary of the results from Figure 5.

6. Details about the clinical sample testing (comparator multiplexed PCR assay) are missing.

7. The manuscript could benefit from additional proofreading and copyediting.

Reviewer #1:

In this manuscript, Xu et al proposed a new microfluidic platform that couples CRISPR/Cas12a and multiplex RPA for the testing of multiple nucleic acids. Considering current pandemic, it should be very meaningful to have such a simple but efficient tool for rapid diagnosis of infectious diseases. This strategy realizes the multiplexed detection relying on the “microfluidic-based space coding”, which utilizes the specific spatial spots to indicate each individual targets after the recognition of relevant crRNA. ... This is useful in real applications to efficiently reduce false positive and false negative results. I think this study is of great impact to the field and also of great interest to Nature Communications’ readers.

Reply: We really appreciate the reviewer’s positive comments on our work.

1. When describe the SS-Chip, “Inlet”, “inlet”, “Outlet” and “outlet” were mixedly used. Please check the whole manuscript and format the use.

Reply: Thank you for this comment. We have made relevant changes accordingly.

2. Why use Aquapel treatment to make the device hydrophobic? This part is not clear. And how to carry out the treatment? Please give more details.

Reply: We appreciate this comment. We have added more details in Supplementary Information (Page 3) to explain why and how to do the Aquapel treatment.

Changes:

4. Aquapel treatment

To perform the on-chip assay, we first load different Detection Master Mixes (DMMs, including Cas12a, different crRNAs, and the reporter) in the relevant outlet wells, and then inject the sample from the central inlet to disperse it into each outlet. To avoid the DMM solution to flow back from the outlet into the inlet and cause unwanted cross contamination during the testing, Aquapel was used to make the microchannels hydrophobic. This ensures the DMM solution to stay in the designated outlet and facilitates the mixing of DMM and the sample.

It is simple to carry out the Aquapel treatment. The treatment procedure is: use a pipette to drop ~1 mL Aquapel onto the surface of the glass slide, spread the Aquapel evenly with the pipette tip, wipe out the extra Aquapel on the glass slide after 10-min drying.

3. *In discussion, it is recommended to add more details to compare the SS-Chip and other microfluidic platforms, which helps the readers further understand the unique advantages of this work.*

Reply: Thanks for this great comment. We have added more discussion about the microfluidics-based strategies for nucleic acid detection and cited several references (Maintext, Page 24).

Changes: However, common microfluidic platforms have complicated fluidic networks, and often require valves, accessory pumps or other complex controlling systems.^{41,42} Thus, they are not ideal for the quick and on-site detection of multiple targets.⁴³ For example, some droplet- and electrochemistry-based microfluidic biosensors were proposed for nucleic acid detection, but the targets that can be analyzed simultaneously are limited.^{44,45} Though CARMEN and mCARMEN enables the testing of over one hundred nucleic acid targets, the relatively complex setup or expensive instrumentation hinders their wide use.

4. *It is interesting to demonstrate the usage of an eye mask for potential point-of-care testing. How to know that the steam mask can heat the SS-Chip at 40 °C for 20 min? It is recommended to find a way to measure the temperature of the mask and the device, and add data to support it.*

Reply: We appreciate this suggestion. We have used an infrared thermometer to measure the temperature of the eye mask surface (the device was wrapped inside). The temperature against the time is plotted as Figure S9b. We can find that the temperature increased from room temperature to about 40°C and then can maintain ~20 min, which is within an acceptable range for the RPA assay in a point-of-care testing. We have added this data into Supplementary Information (Page 14).

Figure S9 Demonstration of MiCaR as a tool for potential point-of-care testing. (a) Heating the SS-Chip by using a steam eye mask. Heat can be produced when the carbonyl iron powder inside the mask is rapidly oxidized after contacting with oxygen in the air. (b) Temperature on the surface of the eye mask measured with an infrared thermometer. The steam mask can heat the SS-Chip at ~40 °C for 20 min. (c) Detection results of Sample #53. The results obtained in this point-of-care way were highly consistent with those obtained on a normal laboratory dry bath.

5. In Figure 3d, e, f, is the heatmap correlated with the gel results? It seems that the intensities do not match between the two groups of results. Please clarify.

Reply: Thanks for this great comment. Figure 3d, e and Figure 3f are two groups of experimental results (In the revised version, these have been labelled as Figure 3c, d, e). The RPA products were generated in different batches. Therefore, the heatmap is not correlated with the gel results. To avoid misunderstanding, we have added a sentence in the caption of Figure 3 (Maintext, Page13).

Changes: To be noted, the amplicons used in c, d and e were from different batches of RPA assays.

6. There are some typos (e.g., Page 26, the PDMS slab to was cut and....) in the manuscript. Please check.

Reply: Thank you very much. We have carefully checked the whole manuscript and made relevant changes.

7. The references are not in the same format.

Reply: Thanks. We have checked all the references and formatted them according to the reference style of Nature Communications.

Reviewer #2:

This manuscript by Xu, Chen, and coworkers describes a microfluidic approach for multiplexed nucleic acid detection using recombinase polymerase amplification (RPA) and Cas12a-based detection. The authors use their approach, MiCar, to develop an assay for the detection of 9 HPV subtypes. The authors then test their assay on a set of 100 patient samples, and demonstrate high concordance with a clinical multiplexed PCR assay. Overall, I appreciate the work the authors have done to multiplex RPA and combine it with Cas12a detection.

Reply: We really appreciate the reviewer's positive comments on our work.

Major comments:

1. What is the benefit of doing multiplexed RPA rather than multiplexed PCR? The authors justify this in the introduction by mentioning that RPA is fast and compatible with one-pot detection, but they do not (to my knowledge) show any one-pot data. The authors should include additional justification for the use of RPA in the introduction.

Reply: We thank the reviewer for this good suggestion. We have made relevant changes to give additional justification for the use of RPA in Introduction (Maintext, Page 4).

Changes: RPA, a fast and high-fidelity amplification method, is used more commonly because the reaction requires only a single temperature (37–42°C) that can be easily achieved with simple heaters and is also compatible the optimum temperature of CRISPR-Cas systems.

2. A key question is the generalizability of the author's approach for designing and optimizing primer sequences for multiplexed RPA. The authors should demonstrate that their approach can be applied to at least one other multiplexed panel.

Reply: Thanks for this great suggestion. We have applied our approach to test a respiratory virus panel (RVP) with eight of the most clinically relevant viruses, which are influenza B virus (FLUBV), human coronavirus NL63 (HCoV-NL63), human coronavirus OC43 (HCoV-OC43), human respiratory syncytial virus (HRSV), human Coronavirus HKU1 (HCoV-HKU1), SARS-CoV-2, human parainfluenza virus serotype 3 (HPIV-3) and human metapneumovirus (HMPV). We performed a series of experiments to demonstrate that our approach also worked well for this multiplexed panel.

1. Based on the basic designing principles, multiple pairs of RPA primers were designed for each of the eight respiratory viruses (as shown in the following Table S9), and the optimum primer was selected out based on the agarose gel electrophoresis (Figure S10, as indicated by the red numbers).

Table S9 RPA primers designed for the 8 respiratory viruses. The optimum primers finally used were shown as bold fonts.

Name	Sequence	Name	Sequence
FLUBV	1 Forward GAACGGGACATAAAGCAACAATAAAGCTCCTTC	HCoV-HKU1	1 Forward CCCAATCATCTGGTGTATTCTCTGAAAAATCC
	Reverse CCTGCTCTTTTCATTGGTATTCTCTTTATG		Reverse CCTGAAATAAACCAATCATCAAGACTTGACTG
	2 Forward CATAAAGAGATACCAATGAAAGTAAAGACAGG		2 Forward CCCAATCATCTGGTGTATTCTCTGAAAAATCC
	Reverse CCTCTTTTCATTCCACCACGGCAAAACAC		Reverse CTGAATAAACCAATCATCAAGACTTGACTG
HCoV-NL63	3 Forward CATAAAGAGAAATACCAATGAAAGTAAAGACAGG	SARS-CoV-2	3 Forward CCAATCATCTGGTGTATTCTCTGAAAAATCC
	Reverse CTGCTCTGTTATTCTTTCAAGTCATAAGCC		Reverse CTACACCTGAATAAACCAATCATCAAGACTTG
	4 Forward CTATGACTGAAAGATAACCGAGAGACAGGCC		3 Forward CTGCTATCCAGTTGGCTACCAAAATGAATGC
	Reverse CCATCCATAAGATTCTCTGTTTCCAAATG		Reverse CATTATGGTATTCGGCAAGACTATGCTCAGGTC
HCoV-OC43	1 Forward CTTGGTATAGTGTAACTTCTTAGTGAAC	HPIV-3	1 Forward CTGCTATCCAGTTGGCTACCAAAATGAATGC
	Reverse CATAATCCAAAACAACCAACCACTTGCTC		Reverse CAGATTATTGTTGTTATTCGGCAAGACTATGC
	2 Forward CAAGTTGGTTGTTTGGATTAGGCC		3 Forward CAGAAGTAGGACCTGAGCATAGTCTTGCCG
	Reverse CAGTATGAAATCAACATCAACAAACAGAC		Reverse CAACATAAGGAAACACACACCTCCAAAGG
HRSV	3 Forward CAAGTTGGTGTGTTTGGATTAGTCC	HMPV	4 Forward CCTGAGCATAGTCTTGCCGAATACCAATG
	Reverse CAGTATGAAATCAACATCAACAAACAGAC		Reverse CAACATAAGGAAACACACACCTCCAAAGG
	1 Forward ATGTTGAGGAGGCGAGGAGGAAAGTTGATAACCC		5 Forward CATAGTCTTGGCGAATACCAATGAATCTGGC
	Reverse TTACACACACTTCTACGCCGAAACAAACCC		Reverse CCTTGGAACTTCTCCAAACACCTGTATGG
HRSV	2 Forward ATGTTGAGGAGGCGAGGAGGAAAGTTGATAACCC	HPIV-3	1 Forward CGAGATGGAACGAATCAAAGATAAAATACGGG
	Reverse GTACACACACTTCTACGCCGAAACAAACCC		Reverse CCATCTCTTTGCTTACTGCTTCTCACTTC
	3 Forward TGTTGAGGAGGCGAGGAGGAAAGTTGATAACCC		2 Forward CGAGATGGAACGAATCAAAGATAAAATACGGG
	Reverse TTACACACACTTCTACGCCGAAACAAACCC		Reverse CAACCATCTCTTTCTTAGCTGTTCTCTGC
HRSV	4 Forward TGTTGAGGAGGCGAGGAGGAAAGTTGATAACCC	HPIV-3	3 Forward CGAGATGGAACGAATCAAAGATAAAATACGGG
	Reverse TTACACACACTTCTACGCCGAAACAAACCC		Reverse CGAAGCAGATTCCTTTCTTAGTCTACTGG
	5 Forward TTTGGAGGCGAGGAGGAAAGTTGATAACCC		4 Forward CTGGGCTTCACTAGTAGAGATTCAAAGAGTGG
	Reverse TTACACACACTTCTACGCCGAAACAAACCC		Reverse TGATTATCTTTATTGCTCTGTCTTAGTGG
HRSV	6 Forward TTTGGAGGCGAGGAGGAAAGTTGATAACCC	HPIV-3	5 Forward CAACCTTCTTACCTGGCAGTTCAGATAC
	Reverse TTACACACACTTCTACGCCGAAACAAACCC		Reverse TGATTATCTTTATTGCTCTGTCTTAGTGG
	7 Forward TTTGGAGGCGAGGAGGAAAGTTGATAACCC		1 Forward CAGTTTCTGCTGACCAAGTTGGCAAGAGGG
	Reverse TTACACACACTTCTACGCCGAAACAAACCC		Reverse CTTGCTCAAAAATTTTCAGTCTCTCAC
HRSV	8 Forward TTTGGAGGCGAGGAGGAAAGTTGATAACCC	HPIV-3	2 Forward CTACTCCAAAATGAAAGGACTGTGAAACAAGG
	Reverse TTACACACACTTCTACGCCGAAACAAACCC		Reverse ACAAGCAACCAAGGCCCAAGAGGAGATAG
	1 Forward CTAGTGAACAAATATCCACACCCAAAGGACCC		3 Forward CTACCCAAATGAGAAGGACTGTGAAAGAAGGGG
	Reverse CATGTTGGGTTGAGTGTTCATAGTGAG		Reverse ACAAGCAACCAAGGCCCAAGAGGAGATAG
HRSV	2 Forward CAAAATCCACACCCAAAGGACCTCATTAAGAG	HPIV-3	4 Forward CTACCCAAATGAGAAGGACTGTGAAAGAAGGGG
	Reverse CATGTTGGGTTGAGTGTTCATAGTGAG		Reverse ACAAGCAACCAAGGCCCAAGAGGAGATAG
	3 Forward CTCATATGAAACAGCTGAAACCAACAGATGAC		5 Forward CCCAATGAGAGGACTGTGAAACAAGGGG
	Reverse GTGATGACTAACAGTAATCTGAGTAAGGG		Reverse ACAAGCAACCAAGGCCCAAGAGGAGATAG
HRSV	4 Forward CTCATATGAAACAGCTGAAACCAACAGATGAC	HPIV-3	1 Forward CTATCTCTCTTGGGGCTCTGGTGTCTGTATC
	Reverse TGTGATGACTAACAGTAATCTGAGTAAGGG		Reverse CAATGATGAAAGCCAGTGTCTCTCTCTGTC
	5 Forward CTGAAACAAGGAGCATTCAATACATAAAGCC		
	Reverse CGTGTAGCTGTGTCTTCCAAATTTGTTGTAAC		
HRSV	6 Forward CAACAAGGAGCATTCAATACATAAAGCC		
	Reverse CGTGTAGCTGTGTCTTCCAAATTTGTTGTAAC		

Figure S10 Evaluation of the RPA primers for the 8 respiratory viruses. Multiple pairs of primers were designed and tested for each of the eight viruses. The primer producing a relatively denser band was selected as the optimum one, labelled as red fonts on the top of the gel images.

2. Next, the crRNA candidates for recognizing the 8 respiratory viruses were designed against the amplification region determined by the optimum RPA primers. Structural prediction and ΔG calculation were performed using the mFold webserver and OligoAnalyser. The optimum crRNA with the proper secondary structure was selected out based on the prediction results (as shown in Table S10). Then the performance of these crRNAs was evaluated by the 8×8 matrix-based activity test.

Table S10 crRNAs designed for recognizing the target sequence of the 8 respiratory viruses.

Name	Sequence
FLUBV crRNA	UAAUUUCUACUAAGUGUAGAUCAACAAAUAGCCAGAUUAG
HCoV-NL63 crRNA	UAAUUUCUACUAAGUGUAGAUAAAAAGGUGAGUGUUGUAAUU
HCoV-OC43 crRNA	UAAUUUCUACUAAGUGUAGAUCCACCACUGCGCAAAGC
HRSV crRNA	UAAUUUCUACUAAGUGUAGAUACAGGGUGUGGUUACAUCAU
HCoV-HKU1 crRNA	UAAUUUCUACUAAGUGUAGAUUAUGGUUUAUCUGUCACACC
SARS-CoV-2 crRNA	UAAUUUCUACUAAGUGUAGAUACUAAAGAAGGUGCCACUAC
HPIV-3 crRNA	UAAUUUCUACUAAGUGUAGAUUAACUCGGGUCACUAUCAAG
HMPV crRNA	UAAUUUCUACUAAGUGUAGAUAGAACAUUGAAAACAGCCAG

Figure S7a Discrimination of the 8 respiratory viruses using the Cas12a-based assay. (-) was used as Control, in which no plasmid was added.

3. Next, we carried out the 8-plexed RPA to amplify the 8 targets with a concentration of 10^{-12} M for each virus plasmid. The amplicons were tested by the Cas12a-based assay, with the results measured by using a microplate reader.

Figure S7b Evaluation of the 8-plexed RPA products based on the Cas12a-based assay. (-) was used as Control, in which no plasmid was added during the RPA assay.

4. Finally, we used MiCaR to measure the products of the 8-plexed RPA assay.

Figure 7c MiCaR-based readout to evaluate the 8-plexed RPA assay. (-) was used as Control, in which no crRNA was added.

In summary, we performed additional experiments to demonstrate that our approach can be successfully applied to a new multiplexed panel with 8 respiratory viruses, which suggests that our approach could be a general strategy for multiplexed detection of nucleic acids.

Accordingly, we have added a new section and Figure 7 in Maintext (Pages 20-21) with additional experimental details and data in Supplementary Information (Figure S10; Table S9, S10) to demonstrate the generalizability of our approach. We also updated the writing in the relevant sections.

Changes:

Testing the generalizability of MiCaR with a respiratory virus panel (RVP)

Lastly, we applied the MiCaR-based approach to the detection of 8 most clinically relevant respiratory viruses.^{33, 34} This RVP includes influenza B virus (FLUBV), human coronavirus NL63 (HCoV-NL63), human coronavirus OC43 (HCoV-OC43), human respiratory syncytial virus (HRSV), human Coronavirus HKU1 (HCoV-HKU1), SARS-CoV-2, human parainfluenza virus serotype 3 (HPIV-3) and human metapneumovirus (HMPV). We first designed multiple pairs of RPA primers for each of the eight viruses (as shown in Table S9), and the optimum primer was selected out based on the agarose gel electrophoresis (Figure S10, as indicated by the red numbers). Next, the crRNAs for recognizing the 8 respiratory viruses were designed against the amplification region determined by the optimum RPA primers (Table 10). The optimum crRNA with the proper secondary structure was selected out based on the software predication results Then the performance of these crRNAs was evaluated by the 8×8

matrix-based activity test. Figure 7a shows that these crRNAs have great specificity against the relevant target. Subsequently, an 8-plexed RPA was performed and the products were verified by using the Cas12a assay. The results in Figure 7b demonstrates that all the target viruses were successfully amplified and recognized. Furthermore, MiCaR was used to verify the 8-plexed RPA amplicons. The original detection images and bar plots are shown in Figure 7c. All these results further proved that our approach could be a versatile strategy for multiplexed NAT.

Figure 7 Testing of the RVP with the MiCaR-based approach. (a) Discrimination of the 8 respiratory viruses using the Cas12a-based assay. (-) was used as Control, in which no plasmid was added. (b) Evaluation of the 8-plexed RPA products based on the Cas12a-based assay. The template was 10^{-12} M plasmid for each virus. (-) was used as Control, in which no plasmid was added during the RPA assay. (c) MiCaR-based readout to evaluate the 8-plexed RPA assay. (-) was used as Control, in which no crRNA was added. To be noted, the amplicons used in b and c were from different batches of RPA assays.

3. How well does the multiplexed RPA amplification work? The authors appear to have demonstrated it using only a single target concentration of $1e-11$ M. This is a very high target concentration (>1 million copies per microliter). The authors need to perform more rigorous analysis of the sensitivity of their method for detecting each of the HPV subtypes on their panel to establish the limit of detection of their assay.

Reply: We appreciate this great suggestion. Following the Reviewer's comment, we have performed a group of experiments to evaluate the sensitivity of the multiplex RPA. We prepared a series of plasmid samples with different concentrations (10^{-12} , 10^{-13} , 10^{-14} , 10^{-15} , 10^{-16} , 10^{-17} , 10^{-18} and 0 M; each sample was a mixture of the 9 HPV subtypes with the relevant concentration). Then a multiplex RPA assay were carried out for each sample. The RPA

products were evaluated with the Cas12a assay. The results were measured on a microplate reader and shown as following. The 9-plexed RPA assay (1× primer for each target) achieved a sensitivity of 10^{-17} - 10^{-18} M for all the targets except HPV-18. The lower sensitivity for HPV18 could result from a relatively lower primer efficiency during the amplification for low-concentration templates. This was improved by using 3× HPV18 primer in the 9-plexed RPA assay, and finally approached to 10^{-18} M.

Figure 4 Sensitivity test of the 9-plexed RPA assay for the 9 HPV subtypes. (a) Scheme showing that a series of samples with different plasmid concentrations of the 9 targets were prepared, amplified and tested. (b) Titration of the plasmids after RPA based on the Cas12a assay. Note that the 9-plexed RPA assay (1× primer for each target) achieved a sensitivity of 10^{-17} - 10^{-18} M for all the targets except HPV-18, while the sensitivity for HPV-18 could also be improved to 10^{-18} M after using 3× HPV-18 primer in the 9-plexed assay. Values represent the mean \pm SD of three independent experiments. *P-value <0.05 , **P-value <0.01 , ***P-value <0.001 , and ****P-value <0.0001 .

We have added the relevant description and Figure 4 in (Maintext, Pages 14-15) and updated the relevant experimental (Maintext, Pages 29-30).

4. The background intensity (in arbitrary units) appears to vary widely between figure panels. For example, in Figure 4e and 4f the background is ~ 90 a.u., in Figure 4g the background is < 30 a.u., whereas in Figure 5xx the background is ~ 50 a.u.. Some of the samples from the dilution series in Figure 4g have fluorescence values in this range. The authors should more

rigorously characterize the amount of background fluorescence in their assay to ensure that this does not influence assay performance.

Reply: Thanks for this great comment. We checked the experimental records and found that the differences between the background of various experiments mainly resulted from the light source of the microscope (the light power for the excitation was not set as the same) during the imaging. Anyway, we have conducted additional experiments and updated the figures.

Figure 5 (e) Kinetics of SS-Chip-based detection of synthetic HPV-16 plasmids (positive, 10 nM; negative, 0). (f) Fluorescence intensities corresponding to the on-chip assay performed with increasing concentrations of the HPV-16 plasmid. Values represent the mean \pm SD of three independent experiments. (g) Analysis of the fluorescence signals of amplified HPV-16 plasmids. Values represent the mean \pm SD of three independent experiments. *P-value < 0.05, **P-value < 0.01, ***P-value < 0.001, and ****P-value < 0.0001.

Minor comments:

1. In the introduction, there are a few inaccurate statements (for example, CARMEN uses Alexa Fluor dyes, not FAM/HEX/TEX/Cy5). Also, the authors do not mention mCARMEN, which allows for microfluidic-based multiplexing using Cas13 and Cas12 without requiring a complex workflow or dye-based color coding (<https://www.nature.com/articles/s41591-022-01734-1>).

Reply: Thanks for these great comments. We have corrected the relevant statements, and also added additional discussion about mCARMEN in Introduction.

Changes: The readouts are obtained as a pool of color codes that result from different ratios of four Alexa Fluor dyes. (Maintext, Pages 4-5)

Indeed, a recent work developed an improved CARMEN system (named microfluidic CARMEN, mCARMEN), which relies on commercially available Fluidigm microfluidics and

instrumentation.³⁰ It could be a promising strategy for the detection of multiple viruses, but the instrumentation setup and the device are relatively expensive and these might compromise its wide application in common labs and source-restricted areas. (Maintext, Page 5)

2. In Figure 2c, why is the amplicon fidelity low for HPV11 and HPV18?

Reply: Thanks for this great comment. We checked the sequencing data carefully and performed the alignment for the Top 5 sequences of all the 9 HPV subtypes. As shown in the following, the relatively low fidelity of HPV11 and HPV18 did not result from the mutation, but from certain sequences with shortened lengths. This could occasionally happen in RPA, because it is well known that amplicons of different lengths can be generated even in a singleplexed RPA reaction. Anyway, the sequences targeted by the crRNAs (indicated by the black box) were in 100% fidelity for all the sequences. We have added a supporting file (Supplementary Data 4) and a sentence to discuss this in Maintext (Page 11).

```

1      10      20      30      40      50      60      70      80      90
HPV-11 CCTTTAGGCGTTGGTGTAGTGGGCATCCATTGCTAAACAAATATGATGATGTAGAAAATAGTGGTGGGTATGGTGGTAATCCCTGGTCAGGATAATA
4 CTTTAGGCGTTGGTGTAGTGGGCATCCATTGCTAAACAAATATGATGATGTAGAAAATAGTGGTGGGTATGGTGGTAATCCCTGGTCAGGATAATA

100     110     120     130     140     150     160     170     180     190
HPV-11 GGGTAAATGTAGGTATGGATTATAAACAAACCAGCTATGTAATGGTGGGCTGTGCTCCACCGTTAGGTGAACATGGGGTAAGGGTACACAATGTTT
4 GGGTAAATGTAGGTATGGATTATAAACAAACCAGCTATGTAATGGTGGGCTGTGCTCCACCGTTAGGTGAACATGGGGTAAGGGTACACAATGTTT

200     210     220     230     240     250     260     270     280     290
HPV-11 AAATACCTCTGTACAAAATGGTGACTGCCCGCCGTTGGAACCTATTACCAGTGTATACAGGATGGGGACATGGTTGATACAGGCTTTGGTGCATG
4 AAATACCTCTGTACAAAATGGTGACTGCCCGCCGTTGGAACCTATTACCAGTGTATACAGGATGGGGACATGGTTGATACAGGCTTTGGTGCATG

300     310     320
HPV-11 AATTTTCAGACTTACAAACCAATAAATCGGATG
4 AATTTTCAGACTTACAAACCAATAAATCGGATG

1      10      20      30      40      50      60      70      80      90
HPV-11 CCTTTAGCGCTTGGTGTAGTGGGCATCCATTGCTAAACAAATATGATGATGTAGAAAATAGTGGTGGGTATGGTGGTAATCCCTGGTCAGGATAATA
5 .....GCGTGGTGTAGTGGGCATCCATTGCTAAACAAATATGATGATGTAGAAAATAGTGGTGGGTATGGTGGTAATCCCTGGTCAGGATAATA

100     110     120     130     140     150     160     170     180     190
HPV-11 GGGTAAATGTAGGTATGGATTATAAACAAACCAGCTATGTAATGGTGGGCTGTGCTCCACCGTTAGGTGAACATGGGGTAAGGGTACACAATGTTT
5 GGGTAAATGTAGGTATGGATTATAAACAAACCAGCTATGTAATGGTGGGCTGTGCTCCACCGTTAGGTGAACATGGGGTAAGGGTACACAATGTTT

200     210     220     230     240     250     260     270     280     290
HPV-11 AAATACCTCTGTACAAAATGGTGACTGCCCGCCGTTGGAACCTATTACCAGTGTATACAGGATGGGGACATGGTTGATACAGGCTTTGGTGCATG
5 AAATACCTCTGTACAAAATGGTGACTGCCCGCCGTTGGAACCTATTACCAGTGTATACAGGATGGGGACATGGTTGATACAGGCTTTGGTGCATG

300     310     320
HPV-11 AATTTTCAGACTTACAAACCAATAAATCGGATG
5 .....

```

Alignment of the top 5 sequences for HPV-11 in the multiplexed PCR products

```

1      10      20      30      40      50      60      70      80      90
HPV-18 CACTGGGCTAAAGGCACCTGCTTGTAAATCGCGTCCTTTATCACAGGGCGATTGCCCGCCCTTAGAACTTAAAAACACAGTTTTGGAAGATGGTGATA
1 CACTGGGCTAAAGGCACCTGCTTGTAAATCGCGTCCTTTATCACAGGGCGATTGCCCGCCCTTAGAACTTAAAAACACAGTTTTGGAAGATGGTGATA

100     110     120     130     140     150     160     170     180     190
HPV-18 TGGTAGATACCTGGATATGGTGGCCATGGACTTTAGTACATTGCAAGATACTAAATGTGAGGTACCATTTGGATATTTGTCAGTCTATTTGTAATAATCC
1 TGGTAGATACCTGGATATGGTGGCCATGGACTTTAGTACATTGCAAGATACTAAATGTGAGGTACCATTTGGATATTTGTCAGTCTATTTGTAATAATCC

200     210     220     230     240     250     260     270     280     290
HPV-18 TGATTAATTAACAATGCTGCAGATCCCTATGGGGATTCCATGTTTTTTGCTTACGGCGTGAGCAGCTTTTTGCTAGGCATTTTGGAATAGAGCA
1 TGATTAATTAACAATGCTGCAGATCCCTATGGGGATTCCATGTTTTTTGCTTACGGCGTGAGCAGCTTTTTGCTAGGCATTTTGGAATAGAGCA

300     310     320     330     340     350     360     370     380
HPV-18 GGTACTATGGGTGACACTGTGCCTCAATCCCTTATATATTAAGGCACAGGTATGCGTGTCCACCTGGCAGCTGTGTATTTCCCTCTCCAAGTG
1 GGTACTATGGGTGACACTGTGCCTCAATCCCTTATATATTAAGGCACAGGTATGCGTGTCCACCTGGCAGCTGTGTATTTCCCTCTCCAAGTG

390     400     410
HPV-18 GCTCTATTGTTACCTCTGACTCCCAGTTG
1 GCTCTATTGTTACCTCTGACTCCCAGTTG

1      10      20      30      40      50      60      70      80      90
HPV-18 CACTGGGCTAAAGGCACCTGCTTGTAAATCGCGTCCTTTATCACAGGGCGATTGCCCGCCCTTAGAACTTAAAAACACAGTTTTGGAAGATGGTGATA
2 CACTGGGCTAAAGGCACCTGCTTGTAAATCGCGTCCTTTATCACAGGGCGATTGCCCGCCCTTAGAACTTAAAAACACAGTTTTGGAAGATGGTGATA

100     110     120     130     140     150     160     170     180     190
HPV-18 TGGTAGATACCTGGATATGGTGGCCATGGACTTTAGTACATTGCAAGATACTAAATGTGAGGTACCATTTGGATATTTGTCAGTCTATTTGTAATAATCC
2 TGGTAGATACCTGGATATGGTGGCCATGGACTTTAGTACATTGCAAGATACTAAATGTGAGGTACCATTTGGATATTTGTCAGTCTATTTGTAATAATCC

200     210     220     230     240     250     260     270     280     290
HPV-18 TGATTAATTAACAATGCTGCAGATCCCTATGGGGATTCCATGTTTTTTGCTTACGGCGTGAGCAGCTTTTTGCTAGGCATTTTGGAATAGAGCA
2 TGATTAATTAACAATGCTGCAGATCCCTATGGGGATTCCATGTTTTTTGCTTACGGCGTGAGCAGCTTTTTGCTAGGCATTTTGGAATAGAGCA

300     310     320     330     340     350     360     370     380
HPV-18 GGTACTATGGGTGACACTGTGCCTCAATCCCTTATATATTAAGGCACAGGTATGCGTGTCCACCTGGCAGCTGTGTATTTCCCTCTCCAAGTG
2 GGTACTATGGGTGACACTGTGCCTCAATCCCTTATATATTAAGGCACAGGTATGCGTGTCCACCTGGCAGCTGTGTATTTCCCTCTCCAAGTG

390     400     410
HPV-18 GCTCTATTGTTACCTCTGACTCCCAGTTG
2 GCTCTATTGTTACCTCTGACTCCCAGTTG

1      10      20      30      40      50      60      70      80      90
HPV-18 CACTGGGCTAAAGGCACCTGCTTGTAAATCGCGTCCTTTATCACAGGGCGATTGCCCGCCCTTAGAACTTAAAAACACAGTTTTGGAAGATGGTGATA
3 CACTGGGCTAAAGGCACCTGCTTGTAAATCGCGTCCTTTATCACAGGGCGATTGCCCGCCCTTAGAACTTAAAAACACAGTTTTGGAAGATGGTGATA

100     110     120     130     140     150     160     170     180     190
HPV-18 TGGTAGATACCTGGATATGGTGGCCATGGACTTTAGTACATTGCAAGATACTAAATGTGAGGTACCATTTGGATATTTGTCAGTCTATTTGTAATAATCC
3 TGGTAGATACCTGGATATGGTGGCCATGGACTTTAGTACATTGCAAGATACTAAATGTGAGGTACCATTTGGATATTTGTCAGTCTATTTGTAATAATCC

200     210     220     230     240     250     260     270     280     290
HPV-18 TGATTAATTAACAATGCTGCAGATCCCTATGGGGATTCCATGTTTTTTGCTTACGGCGTGAGCAGCTTTTTGCTAGGCATTTTGGAATAGAGCA
3 TGATTAATTAACAATGCTGCAGATCCCTATGGGGATTCCATGTTTTTTGCTTACGGCGTGAGCAGCTTTTTGCTAGGCATTTTGGAATAGAGCA

300     310     320     330     340     350     360     370     380
HPV-18 GGTACTATGGGTGACACTGTGCCTCAATCCCTTATATATTAAGGCACAGGTATGCGTGTCCACCTGGCAGCTGTGTATTTCCCTCTCCAAGTG
3 GGTACTATGGGTGACACTGTGCCTCAATCCCTTATATATTAAGGCACAGGTATGCGTGTCCACCTGGCAGCTGTGTATTTCCCTCTCCAAGTG

390     400     410
HPV-18 GCTCTATTGTTACCTCTGACTCCCAGTTG
3 GCTCTATTGTTACCTCTGACTCCCAGTTG

```

Alignment of the top 5 sequences for HPV-18 in the multiplexed PCR products

Changes: The top 5% sequences of the products showed high-fidelity to the original sequences (Supplementary Data 4). The relatively lower fidelity of HPV-11 and HPV-18 was found resulting from certain amplicons with shortened lengths. Anyway, 100% fidelity was obtained for the crRNA binding region for all 9 targets. (Maintext, Page 11)

3. Figure 3a is confusing, and should be redrawn to make the author's approach clearer.

Reply: Thanks for this great comment. We have redrawn Figure 3a.

Figure 3a Matrix-based reactivity test of the 9 crRNAs against the 9 HPV subtypes.

4. I am confused by the results shown in Figure 4e - for some reason the “negative” sample appears to be increasing in fluorescence over time whereas the “positive” one remains flat.

Reply: Thanks for pointing out this. We made a mistake here and labeled the “negative” and “positive” sample reversely. We have updated Figure 4e.

Figure 4e Kinetics of SS-Chip-based detection of synthetic HPV-16 plasmids (positive, 10 nM; negative, 0).

5. Figure 6 appears to be a summary of the results from Figure 5.

Reply: Thanks for this comment. We have combined previous Figure 5 and Figure 6 as Figure 6, and moved some data to Supplementary Information as Figure S8.

Figure 6 Testing results of 100 patient samples for HPV infection. (a) Brief schematic showing the differences between MiCaR and the clinical assay. The clinical laboratory conducting HPV subtyping assays using multiplexed PCR for target amplification and color coding for detection readouts. Meanwhile, in MiCaR, multiplex RPA was used for amplification, and microchip-based space coding was used for detection readouts. (b) Testing results of Sample #53, which was triple-positive for HPV subtypes, presented as a circle showing the original fluorescence images of the 30 outlet wells. (c) Quantitative analysis of results for Sample #53, tested using MiCaR and the clinical assay. (d) Parallel heat maps showing the results of the 100 samples based on the clinical assay and MiCaR-based testing. Each sample was tested in triplicate using MiCaR. The red circles indicate the inconsistencies (Sample #38 and #77) between MiCaR and the clinical assay. (e) Positive predictive agreement (PPA), negative predictive agreement (NPA), sensitivity and specificity of MiCaR for the detection of the 9 HPV subtypes in clinical samples.

Figure S8 Testing patient samples on MiCaR. (a) Typical on-chip detection results shown with the original images arranged in a circle. To be noted, #4, #45, #50, #83, and #95 are positive samples, and #7, #47, #64, #71, and #96 are negative samples. These results were in consistent with those obtained in clinic. (b) Detailed testing results for each HPV subtype obtained using MiCaR and the clinical laboratory assay.

6. Details about the clinical sample testing (comparator multiplexed PCR assay) are missing.

Reply: We appreciate this suggestion. We have added the information about the clinical sampling testing in Supplementary Information (Page 4).

Changes: Clinical sample testing based on multiplex PCR

The samples were screened in the clinical laboratory for HPV infection by multiplex PCR assays (Tellgen Corporation, Shanghai, China) prior to our assay. Briefly, the HPV DNA released from the sample was amplified by multiplex PCR with biotin-labeled primers. Then the amplicons were hybridized to color-coded microspheres coated with HPV subtype-specific probes. Next, the microspheres were incubated with phycoerythrin (PE)-conjugated streptavidin (SA-PE). After through wash, the microspheres were read on a Luminex 200 system (Luminex Corporation, Texas, USA). The HPV subtypes were determined based on the fluorescent dye signature carried by the microspheres.

7. The manuscript could benefit from additional proofreading and copyediting

Reply: Thanks for this great comment. We have carefully checked the whole manuscript and made relevant changes.

REVIEWERS' COMMENTS

Reviewer #1 (Remarks to the Author):

The manuscript has been well revised according to my comments, and could be accepted for further publication in Nat Comm.

Reviewer #2 (Remarks to the Author):

The revised version of the manuscript has largely addressed my major concerns. I have only a few small suggestions:

- 1) The manuscript mentions "common labs and source-restricted areas" - do the authors mean "resource-restricted areas"? As written, it's a bit confusing.
- 2) The added text in lines 205-208 could benefit from some further clarification / proofreading.
- 3) Influenza A virus is not part of the author's respiratory virus panel - is there a reason for this?

Reviewer #2:

1. *The manuscript mentions "common labs and source-restricted areas" - do the authors mean "resource-restricted areas"? As written, it's a bit confusing.*

Reply: Thank you for this comment. We have changed this statement (Maintext, Page 5).

Changes: It could be a promising strategy for the detection of multiple viruses, but the instrumentation setup and the device are relatively expensive and these might compromise its wide application in source-restricted areas.

2. *The added text in lines 205-208 could benefit from some further clarification / proofreading.*

Reply: We appreciate this comment. We have rephrased these sentences (Maintext, Page 11).

Changes: The top 5 sequences of the products showed high-fidelity to the original sequences (Supplementary Data 4). The relatively lower amplicon fidelity of HPV-11 and HPV-18 was resulting from one or two amplicons with shorter lengths comparing to the original templates. Anyway, all these amplicons showed a 100% fidelity in the crRNA binding regions.

3. *Influenza A virus is not part of the author's respiratory virus panel - is there a reason for this?*

[Redacted]